# Hyperphosphatemia increases inflammation to exacerbate anemia and skeletal muscle wasting independently of FGF23-FGFR4 signaling

Brian Czaya[1,2], Kylie Heitman[1], Isaac Campos[1], Christopher Yanucil[1], Dominik Kentrup[1], David Westbrook[1], Orlando Gutierrez[1], Jodie L Babitt[3], Grace Jung[2], Isidro B Salusky[4], Mark Hanudel[4], Christian Faul[1]*

[1]Division of Nephrology and Hypertension, Department of Medicine, The University of Alabama at Birmingham, Birmingham, United States; [2]Department of Medicine, David Geffen School of Medicine at UCLA, Los Angeles, United States; [3]Division of Nephrology, Program in Membrane Biology, Massachusetts General Hospital, Harvard Medical School, Boston, United States; [4]Department of Pediatrics, David Geffen School of Medicine at UCLA, Los Angeles, United States

**Abstract** Elevations in plasma phosphate concentrations (hyperphosphatemia) occur in chronic kidney disease (CKD), in certain genetic disorders, and following the intake of a phosphate-rich diet. Whether hyperphosphatemia and/or associated changes in metabolic regulators, including elevations of fibroblast growth factor 23 (FGF23) directly contribute to specific complications of CKD is uncertain. Here, we report that similar to patients with CKD, mice with adenine-induced CKD develop inflammation, anemia, and skeletal muscle wasting. These complications are also observed in mice fed high phosphate diet even without CKD. Ablation of pathologic FGF23-FGFR4 signaling did not protect mice on an increased phosphate diet or mice with adenine-induced CKD from these sequelae. However, low phosphate diet ameliorated anemia and skeletal muscle wasting in a genetic mouse model of CKD. Our mechanistic in vitro studies indicate that phosphate elevations induce inflammatory signaling and increase hepcidin expression in hepatocytes, a potential causative link between hyperphosphatemia, anemia, and skeletal muscle dysfunction. Our study suggests that high phosphate intake, as caused by the consumption of processed food, may have harmful effects irrespective of pre-existing kidney injury, supporting not only the clinical utility of treating hyperphosphatemia in CKD patients but also arguing for limiting phosphate intake in healthy individuals.

## Editor's evaluation

Many of us have followed the work of your group and others on FGF23 excess and hyperphosphatemia and the morbidity of chronic kidney disease in both animal models and in the human disease state. Inflammation, anemia and muscle wasting are clearcut serious consequences to deal with CKD. Animal models that will allow us to further elucidate the signaling and effector pathways activated by hyperphosphatemia are welcome advances in the field.

## Introduction

Phosphate (Pi) is an essential mineral nutrient (*Erem and Razzaque, 2018*). Once absorbed and in circulation, Pi is utilized by cells for various structures and functions. Pi metabolism is regulated by

**\*For correspondence:**
cfaul@uabmc.edu

a specific set of hormones to maintain physiological Pi concentrations. Fibroblast growth factor 23 (FGF23) is the chief hormone maintaining body Pi balance by promoting renal Pi excretion when Pi load is high (*Fukumoto and Yamashita, 2007*; *Isakova et al., 2011*). Dysregulation of this system causes either low (hypophosphatemia) or high (hyperphosphatemia) serum Pi levels (*Farrow et al., 2011*; *White et al., 2001*; *Wolf, 2012*). Hyperphosphatemic states can result from various conditions, including rare genetic disorders, such as familial tumoral calcinosis (FTC), and acquired diseases, such as chronic kidney disease (CKD), which are more frequent. Moreover, increased consumption of foods and drinks rich in Pi-based additives is expanding in Westernized diets, leading to excess dietary Pi intake (*Carrigan et al., 2014*; *Gutiérrez et al., 2010a*).

CKD patients have an increased risk of death that is attributable to complications such as inflammation, anemia, and skeletal muscle wasting (*Amdur et al., 2016*; *Hoshino et al., 2020*; *Stenvinkel et al., 2016*). The etiology of CKD-associated anemia is multifactorial and includes absolute iron deficiency and functional iron deficiency, with the latter caused by inflammatory cytokines including interleukin-6 (IL6) and interleukin-1β (IL1β). These inflammatory mediators can directly induce the release of the liver-hormone hepcidin, the master regulator of iron metabolism (*Verga Falzacappa et al., 2007*; *Ganz and Nemeth, 2012*; *Kanamori et al., 2017*). Hepcidin controls the flow of iron into circulation by regulating the iron exporter ferroportin (FPN) (*Nemeth et al., 2004b*). Hepcidin binding occludes FPN (*Aschemeyer et al., 2018*; *Billesbølle et al., 2020*) and induces its degradation, thereby restricting iron efflux into the circulation from iron recycling macrophages, a process also known as reticuloendothelial system (RES) blockade, and from duodenal enterocytes responsible for dietary iron absorption. Collectively, these events reduce serum iron levels (hypoferremia), limiting the supply of iron for erythrocyte production (*Nemeth et al., 2004a*).

Inflammatory cytokines such as IL6 and IL1β also act on skeletal muscle and induce muscle wasting, a comorbidity affecting 65% of CKD patients (*Kovesdy et al., 2013*; *Li et al., 2009*; *Zhang et al., 2013*). The loss of protein from muscle is ascribed to protein degradation by the ubiquitin–proteasome system, suppression of protein synthesis and impaired growth of new muscle fibers (*Wang and Mitch, 2014*). As pleotropic activities of IL6 and IL1β induce the production of myostatin, a pivotal mediator of skeletal muscle wasting, these actions foster the simultaneous induction of atrophy-related gene programs and reduced cellular responses to progrowth signals, which initiates protein synthesis suppression. Together, inflammation and myostatin advance these CKD-associated comorbidities which reduce the survival and quality of life of CKD patients (*Zhang et al., 2011*).

A prominent aspect of CKD is altered mineral metabolism, where hyperphosphatemia and excess serum FGF23 are factors associated with inflammation, anemia, and mortality (*Mehta et al., 2017*; *Munoz Mendoza et al., 2017*; *Navarro-González et al., 2009*; *Tran et al., 2016*). Drugs have been developed to control hyperphosphatemia, but reports show conflicting results about outcomes. Studies of the effects of dietary Pi restriction in animal models are scarce. Dietary interventions to lower Pi intake are challenging because they require long-term behavioral changes made more difficult by the lack of disclosure of Pi content of foods and beverages by the food industry (*Gutiérrez and Wolf, 2010b*).

All cell types rely on Pi for housekeeping roles, and metabolic Pi uptake is facilitated by three families of sodium–Pi (Na/Pi) cotransporters. Type III Na/Pi cotransporters, PiT-1 and PiT-2, are ubiquitously expressed and mediate cellular Pi homeostasis in all cells (*Lederer and Miyamoto, 2012*). Pathologic Pi accumulation in vasculature is mediated by PiT-1 and PiT-2, loading Pi into vascular smooth muscle cells where it activates signaling networks such as Ras/mitogen-activated protein kinase (MAPK) and nuclear factor kappa-light-chain-enhancer of activated B cells (NFκB) (*Chavkin et al., 2015*; *Turner et al., 2020*; *Zhao et al., 2011*). These pathways provide plausible pathomechanisms that support excess Pi as a potential culprit behind the clinical association between hyperphosphatemia and CKD-associated vascular calcification.

Under physiologic conditions, bone-derived FGF23 targets the kidney to increase Pi excretion by activating its canonical signaling complex, fibroblast growth factor receptor 1 (FGFR1) and co-receptor Klotho (*Czaya and Faul, 2019a*). When FGF23 is in pathological excess, as found in dietary Pi overload or CKD, increased FGF23 targets the heart and liver by activating FGFR4, independently of Klotho (*Faul et al., 2011*; *Grabner et al., 2015*; *Singh et al., 2016*). This noncanonical mechanism recruits FGFR4 as a pathologic receptor mediating the effects of excess FGF23 to cause cardiac hypertrophy and promote inflammation (*Faul et al., 2011*; *Grabner et al., 2015*; *Han et al., 2020*;

*Leifheit-Nestler et al., 2017*; *Singh et al., 2016*; *Xiao et al., 2019*). However, whether excess FGF23 and/or Pi directly contribute to functional iron deficiency or skeletal muscle wasting is unknown, and direct actions of excess Pi on the liver have not been studied to date.

In this study, we examine whether hyperphosphatemia and/or pathologic FGF23-FGFR4 signaling aggravates functional iron deficiency and skeletal muscle wasting, the common comorbidities in CKD. We expose mice constitutively lacking FGFR4 to hyperphosphatemia in the absence and presence of CKD. To further define the contribution of hyperphosphatemia, we subject Alport mice, a genetic model of progressive CKD, to a low Pi diet treatment. We identify a molecular mechanism in cultured mouse primary hepatocytes that links excess Pi to its actions on inflammation and iron metabolism, as increased inflammatory cytokines promote widespread pathologies and hepcidin production. Our findings reveal additional complications of hyperphosphatemia besides vascular calcification and identify plausible pathomechanisms underlying clinical associations between inflammation, anemia, and skeletal muscle wasting, which could be targeted therapeutically.

## Results

### FGF23-FGFR4 signaling does not contribute to functional iron deficiency in adenine-induced CKD

To examine FGF23 inflammatory actions in vivo, we explored pathologic FGF23-FGFR4 signaling and its role in functional iron deficiency. We subjected wild-type (*Fgfr4*$^{+/+}$) and constitutive FGFR4 knockout (*Fgfr4*$^{-/-}$) mice to adenine diet to induce CKD. Following adenine diet for 14 weeks, *Fgfr4*$^{+/+}$ and *Fgfr4*$^{-/-}$ mice displayed comparable renal dysfunction, as shown by elevations in blood urea nitrogen (BUN) and serum creatinine levels (*Figure 1A*). On adenine diet, serum FGF23 and Pi levels significantly increased in both genotypes (*Figure 1B*), as expected with marked kidney injury. No significant changes in serum calcium levels were observed between genotypes on adenine (*Figure 1—figure supplement 1A*), despite increased serum Pi levels (*Figure 1B*).

We assessed gene expression of inflammatory cytokines and acute phase proteins in both genotypes in adenine-induced CKD. Unlike in healthy control mice, liver IL1β (*Il1b*), IL6 (*Il6*), and serum amyloid A1 (*Saa1*) transcript levels were significantly and similarly elevated (*Figure 1C, D*) in both *Fgfr4*$^{+/+}$ and *Fgfr4*$^{-/-}$ mice.

To identify if FGF23-FGFR4 signaling contributes to functional iron deficiency, we evaluated liver hepcidin (*Hamp*) mRNA and hematological responses. Compared with control mice, *Hamp* transcript levels were significantly and similarly elevated in both *Fgfr4*$^{+/+}$ and *Fgfr4*$^{-/-}$ mice on adenine (*Figure 1E*). Complete blood count and serum analyses displayed significant reductions in red blood cell (RBC) count, mean corpuscular volume (MCV), hemoglobin, and serum iron levels on adenine diet (*Figure 1F*). Significant reductions in hematocrit percentage (HCT%), mean corpuscular hematocrit (MCH), and serum transferrin saturation percentage (TSAT%) were also observed (*Figure 1—figure supplement 1B*). Spleen tissue sections stained with Perls' Prussian blue revealed profound intracellular iron sequestration, indicating severe RES blockade in both *Fgfr4*$^{+/+}$ and *Fgfr4*$^{-/-}$ mice on adenine (*Figure 1G*, *Figure 1—figure supplement 1C*). Taken together, our data demonstrate that FGF23-FGFR4 signaling does not affect inflammation, the acute phase response or functional iron deficiency and anemia in adenine-induced CKD.

### FGF23-FGFR4 signaling does not contribute to hypoferremia following dietary Pi overload

Excess Pi is a hallmark of CKD, but direct pathologic effects of Pi on tissues other than the vasculature are poorly understood (*Komaba and Fukagawa, 2016*; *Scialla and Wolf, 2014*). To examine if liver Pi deposition is increased in CKD, we analyzed liver Pi levels in adenine-induced mouse CKD model by colorimetric quantification. Hepatic Pi levels were elevated in both *Fgfr4*$^{-/-}$ and *Fgfr4*$^{+/+}$ mice following adenine (*Figure 1H*) but less so in *Fgfr4*$^{-/-}$ mice when compared to *Fgfr4*$^{+/+}$ mice.

To establish whether excess Pi and/or FGF23 contributes to hypoferremia in the absence of CKD, we exposed *Fgfr4*$^{+/+}$ and *Fgfr4*$^{-/-}$ mice to a graded dietary Pi load for 12 weeks. Serum FGF23 levels increased in both genotypes on 2% Pi and 3% Pi diet, in comparison to mice on 0.7% Pi diet (*Figure 2A*). Despite 2% Pi increasing serum FGF23, serum Pi levels significantly increased only on 3% Pi, in comparison to mice fed 0.7% Pi (*Figure 2A*). Notably, these serum Pi levels are comparable

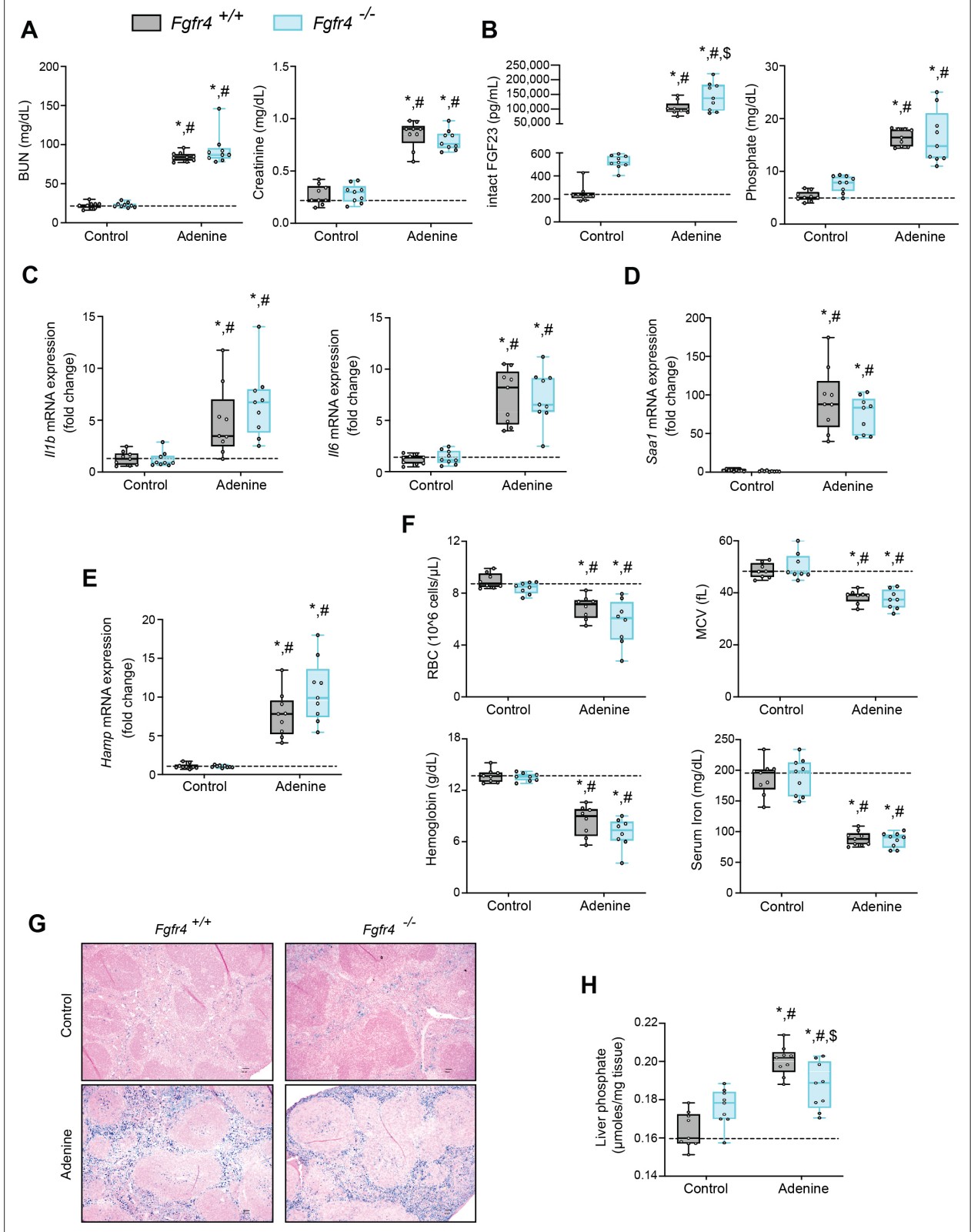

**Figure 1.** FGF23-FGFR4 signaling does not contribute to functional iron deficiency in adenine-induced CKD. Blood urea nitrogen (BUN), serum creatinine (**A**), serum FGF23 and serum phosphate (Pi) levels (**B**). Quantitative polymerase chain reaction (qPCR) analysis of *Il1b*, *Il6*, *Saa1* (**C, D**) and *Hamp* (**E**) expression levels in liver tissue. (**F**) Complete blood count (CBC) analysis. (**G**) Representative gross pathology of Perls' Prussian blue-stained spleen sections (scale bar, 50 µm). Larger magnification is shown in supplementary figure and legends. (**H**) Liver Pi levels. All values are mean ± standard

*Figure 1 continued on next page*

*Figure 1 continued*

error of the mean (SEM; *n* = 8–9 mice/group; *p ≤ 0.05 vs. *Fgfr4*$^{+/+}$ + control diet, #p ≤ 0.05 vs. *Fgfr4*$^{-/-}$ + control diet, $p ≤ 0.05 vs. *Fgfr4*$^{+/+}$ + adenine diet) where statistical analyses were calculated by two-way analysis of variance (ANOVA) followed by Tukey's multiple comparison post hoc test. Dotted lines indicate median *Fgfr4*$^{+/+}$ + control diet measurements.

The online version of this article includes the following figure supplement(s) for figure 1:

**Figure supplement 1.** FGF23-FGFR4 signaling does not contribute to functional iron deficiency in adenine-induced CKD.

to the elevated serum Pi levels observed in adenine-induced CKD (*Figure 1B*). No significant differences in serum calcium levels were observed between genotypes (*Figure 2—figure supplement 1A*), despite elevated Pi levels (*Figure 2A*). No pathologic changes were detected in kidneys regardless of genotype as BUN, serum creatinine, and kidney tissue sections stained with hematoxylin and eosin (H&E) appeared similar to those of mice on 0.7% Pi (*Figure 2—figure supplement 1B–D*). Interstitial fibrosis was not detected in kidneys as shown by Masson's trichrome staining (*Figure 2—figure supplement 1E*). These data indicate elevations in serum levels of FGF23 and Pi in mice on 2% Pi or 3% Pi diet are consequences of an increasing dietary Pi load and not renal dysfunction.

As a high Pi diet has been reported to exacerbate inflammation and serum FGF23 levels (*Sugihara et al., 2017*; *Takashi et al., 2019*; *Yamada et al., 2014*), we evaluated gene expression of inflammatory cytokines and acute phase proteins. Compared to 0.7% Pi, liver *Il1b*, *Il6*, and *Saa1* transcript levels significantly increased in both *Fgfr4*$^{+/+}$ and *Fgfr4*$^{-/-}$ mice on 3% Pi, although not on 2% Pi diets (*Figure 2B, C*). Liver injury was not detected, as no significant elevations in hepatic alanine aminotransferase (*Alt1*) or aspartate aminotransferase (*Ast1*) mRNA levels were found on 3% Pi (*Figure 2—figure supplement 2A, B*). These data support the notion that dietary Pi overload induces inflammation, but not via FGF23-FGFR4 signaling.

To explain these effects of 3% Pi diet and determine if increased tissue Pi deposition is associated with adverse outcomes, we analyzed the relationship between liver and serum Pi levels in both *Fgfr4*$^{+/+}$ and *Fgfr4*$^{-/-}$ mice following a graded dietary Pi load. A positive correlation was detected between hepatic and serum Pi levels in both genotypes, beginning with 2% Pi (*Figure 2D*). These results show liver Pi deposits increase following elevations in dietary Pi content.

Next, we tested if increased liver Pi accumulation affects correlations between liver Pi and liver *Hamp* mRNA levels, as a high Pi diet induces *Hamp* expression (*Nakao et al., 2015*). A positive correlation between liver Pi and liver *Hamp* mRNA levels were detected in both *Fgfr4*$^{+/+}$ and *Fgfr4*$^{-/-}$ mice, again only with diets containing 2% or 3% Pi (*Figure 2E*). As liver injury was not detected (*Figure 2—figure supplement 2A, B*), these data indicate that elevations in liver *Hamp* mRNA are independent of liver injury and may result from Pi-driven inflammation.

We next explored if increased dietary Pi loading led to changes in hematological responses. Marked reductions in RBC, MCV, hemoglobin, and serum iron levels were detected on 3% Pi and were similar in both *Fgfr4*$^{+/+}$ and *Fgfr4*$^{-/-}$ mice (*Figure 2F*). HCT% and MCH were also significantly decreased (*Figure 2—figure supplement 2C*). Supporting these findings, spleen tissue sections revealed that both *Fgfr4*$^{+/+}$ and *Fgfr4*$^{-/-}$ mice showed increased intracellular iron deposits on 3% Pi (*Figure 2G*, *Figure 2—figure supplement 1F*). Thus, dietary Pi loading causes iron restriction and hypoferremia even in the absence of CKD.

## Mouse models of hyperphosphatemia exhibit signs of skeletal muscle wasting which are independent of FGF23-FGFR4 signaling

As inflammation is a known contributor of muscle wasting (*Raj et al., 2008*; *Schaap et al., 2006*; *Verzola et al., 2017*), and since hyperphosphatemia and excess FGF23 are associated with inflammation, we explored whether hyperphosphatemia contributes to skeletal muscle wasting, and if pathologic FGF23-FGFR4 signaling is involved in these effects. We analyzed skeletal muscle from *Fgfr4*$^{+/+}$ and *Fgfr4*$^{-/-}$ mice exposed to adenine-induced CKD (*Figure 1*) or a graded dietary Pi load (*Figure 2*). Examination of skeletal muscle strength indicates that mice on adenine or 3% Pi diet exhibit reduced grip strength in both *Fgfr4*$^{+/+}$ and *Fgfr4*$^{-/-}$ mice, in comparison to respective control mice (*Figure 3A*). A reduction in gastrocnemius mass was also observed (*Figure 3B*). Notably, gastrocnemius metallothionein-1 (*Mt1*) transcript levels were significantly elevated in both genotypes following adenine and

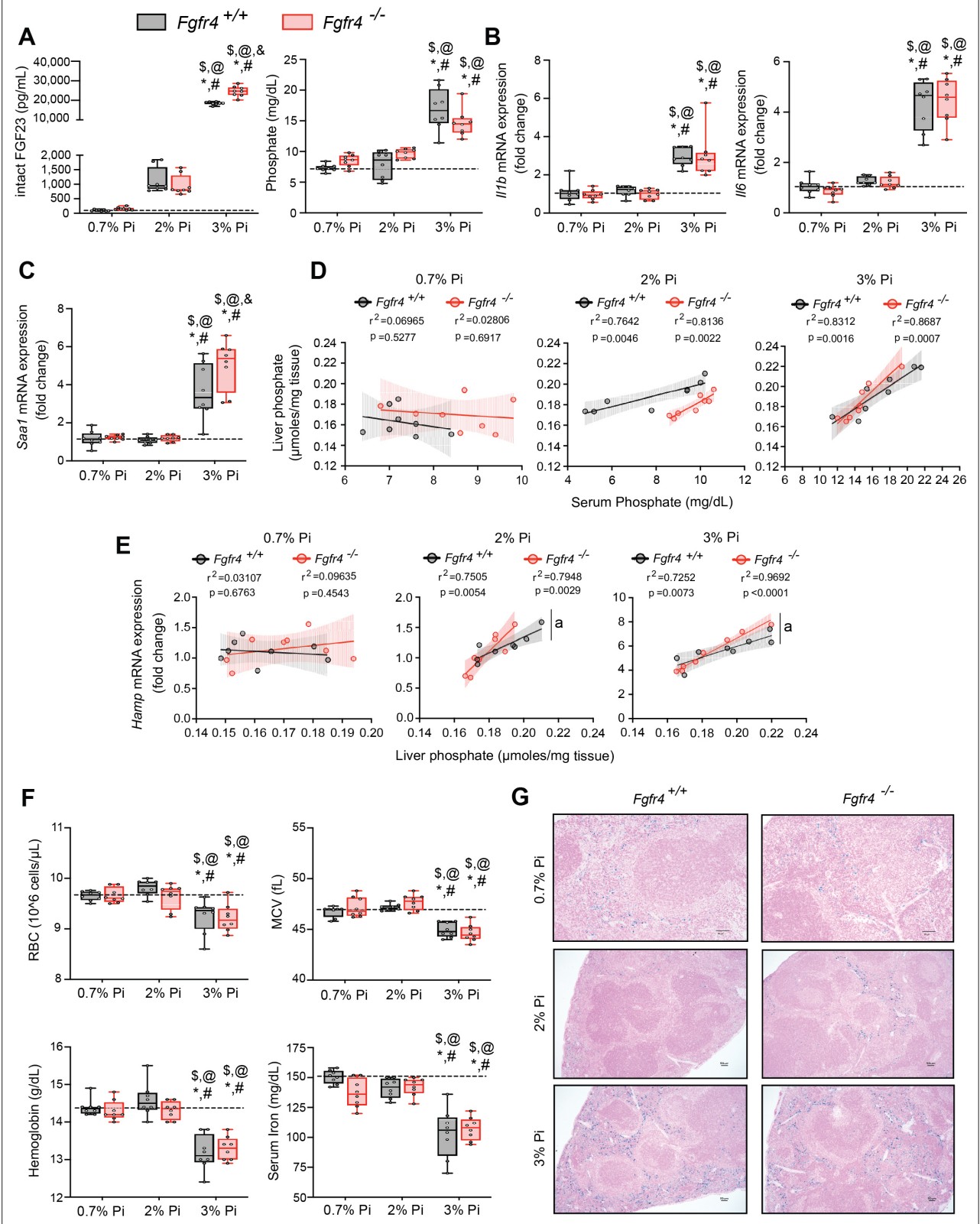

**Figure 2.** FGF23-FGFR4 signaling does not contribute to hypoferremia following dietary Pi overload. (**A**) Serum FGF23 and serum Pi levels. (**B, C**) Quantitative polymerase chain reaction (qPCR) analysis of *Il1b*, *Il6*, and *Saa1* expression levels in liver tissue. (**D**) Scatter plots showing correlations between liver Pi and serum Pi levels. (**E**) Scatter plots showing correlations between liver *Hamp* expression and liver Pi levels (a = slopes are significantly different from each other). (**F**) CBC analysis. (**G**) Representative gross pathology of Perls' Prussian blue-stained spleen sections (scale bar, 50 µm). Larger

*Figure 2 continued on next page*

*Figure 2 continued*

magnification is shown in supplementary figure and legends. All values are mean ± standard error of the mean (SEM; $n$ = 8 mice/group; *p ≤ 0.05 vs. $Fgfr4^{+/+}$ + 0.7% Pi diet, #p ≤ 0.05 vs. $Fgfr4^{-/-}$ + 0.7% Pi diet, \$p ≤ 0.05 vs. $Fgfr4^{+/+}$ + 2% Pi diet, @p ≤ 0.05 vs. $Fgfr4^{-/-}$ + 2% Pi diet, &p ≤ 0.05 vs. $Fgfr4^{+/+}$ + 3% Pi diet) where statistical analyses were calculated by two-way analysis of variance (ANOVA) followed by Tukey's multiple comparison post hoc test. Dotted lines indicate median $Fgfr4^{+/+}$ + 0.7% Pi diet measurements. Scatter plot shadows indicate 95% confidence interval.

The online version of this article includes the following figure supplement(s) for figure 2:

**Figure supplement 1.** FGF23-FGFR4 signaling does not contribute to hypoferremia following dietary Pi overload.

**Figure supplement 2.** Liver injury marker and hematological analyses in $Fgfr4^{+/+}$ and $Fgfr4^{-/-}$ mice fed a graded Pi diet.

3% Pi diets (*Figure 3—figure supplement 1A, B*), indicating that either condition fosters skeletal muscle abnormalities.

We next investigated if these muscle deficits resulted from inflammation inducing myostatin and downstream atrophy-related gene programs, as both experimental models display elevated levels of liver *Il1b* and *Il6* (*Figures 1C and 2B*). Compared to respective control mice, gastrocnemius myostatin (*Mstn*) transcript levels were significantly elevated in both $Fgfr4^{+/+}$ and $Fgfr4^{-/-}$ mice following adenine or 3% Pi diet (*Figure 3C*). Additionally, both genotypes on 2% Pi showed an increased trend in *Mstn* mRNA levels (*Figure 3C*). As these findings suggest increased myofibrillar protein degradation, we further analyzed the expression of two specific ubiquitin ligases of muscle-protein breakdown, muscle RING-finger protein 1 (*Murf1*) and *Atrogin-1*. Compared with their respective control mice, gastrocnemius *Murf1* and *Atrogin1* transcript levels were significantly elevated in both $Fgfr4^{+/+}$ and $Fgfr4^{-/-}$ mice following adenine and 3% Pi diets (*Figure 3D*).

As elevated myostatin and increased myofibrillar protein degradation are features of skeletal muscle wasting, we assessed if these results prompt a shift toward smaller myofibers. Indeed, gastrocnemius tissue sections stained with H&E from $Fgfr4^{+/+}$ and $Fgfr4^{-/-}$ mice, on either adenine or 3% Pi diet, showed smaller muscle fiber size compared with controls (*Figure 3E*). Taken together, these data suggest skeletal muscle wasting in adenine-induced CKD and hyperphosphatemia does not require FGF23-FGFR4 signaling.

## Low Pi feeding limits functional iron deficiency in *Col4a3−/−* (Alport ) mice

Alport (*Col4a3^{-/-}*) mice are an established model of progressive CKD which develop hyperphosphatemia with severe inflammation, hypoferremia, and anemia (*Francis et al., 2019*). To test if hyperphosphatemia aggravates these pathologic complications, we exposed *Col4a3^{-/-}* mice to a low Pi diet treatment (0.2% Pi) for 6 weeks. In comparison to wild-type (*Col4a3^{+/+}*) mice on normal diet (0.6% Pi), *Col4a3^{-/-}* mice showed renal dysfunction by increased BUN and serum creatinine levels (*Figure 4A*). *Col4a3^{-/-}* mice on low Pi diet displayed a reduction in both parameters (*Figure 4A*), along with reduced pathologic alterations in kidney morphology (*Figure 4—figure supplement 1A*). As compared to wild-type mice, serum levels of FGF23 and Pi significantly increased in *Col4a3^{-/-}* mice on normal diet (*Figure 4B*), but less so in *Col4a3^{-/-}* mice on low Pi diet (*Figure 4B*).

To identify if a low Pi diet affects inflammation or the acute phase response in Alport mice, we assessed gene expression of inflammatory cytokines and acute phase proteins. Compared to wild-type mice on normal diet, liver *Il1b*, *Il6*, and *Saa1* transcript levels were significantly elevated in *Col4a3^{-/-}* mice but much less elevated on 0.2% Pi (*Figure 4C, D*). These data support the notion that excess Pi in Alport mice aggravates inflammation.

As our results show that a low Pi diet decreases inflammation, we next explored its impact on functional iron deficiency. Compared to wild-type mice on normal diet, liver *Hamp* transcript levels were significantly elevated in *Col4a3^{-/-}* mice with complete reversal by low Pi diet (*Figure 4E*). Assessing hematological responses, *Col4a3^{-/-}* mice on normal diet were anemic, with significant reductions in RBC count, MCV, hemoglobin, and serum iron levels (*Figure 4F*), as well as in HCT%, MCH, and TSAT% (*Figure 4—figure supplement 1B*). *Col4a3^{-/-}* mice on normal diet displayed profound intracellular iron sequestration in spleen (*Figure 4G*, *Figure 4—figure supplement 1E*), along with excessive spleen and liver nonheme iron levels (*Figure 4—figure supplement 1C, D*). These effects were substantially ameliorated by low Pi diet in *Col4a3^{-/-}* mice, with improved hematologic parameters and reduced iron deposits in spleen (*Figure 4F, G*). Nonheme iron levels in spleen and liver were reduced

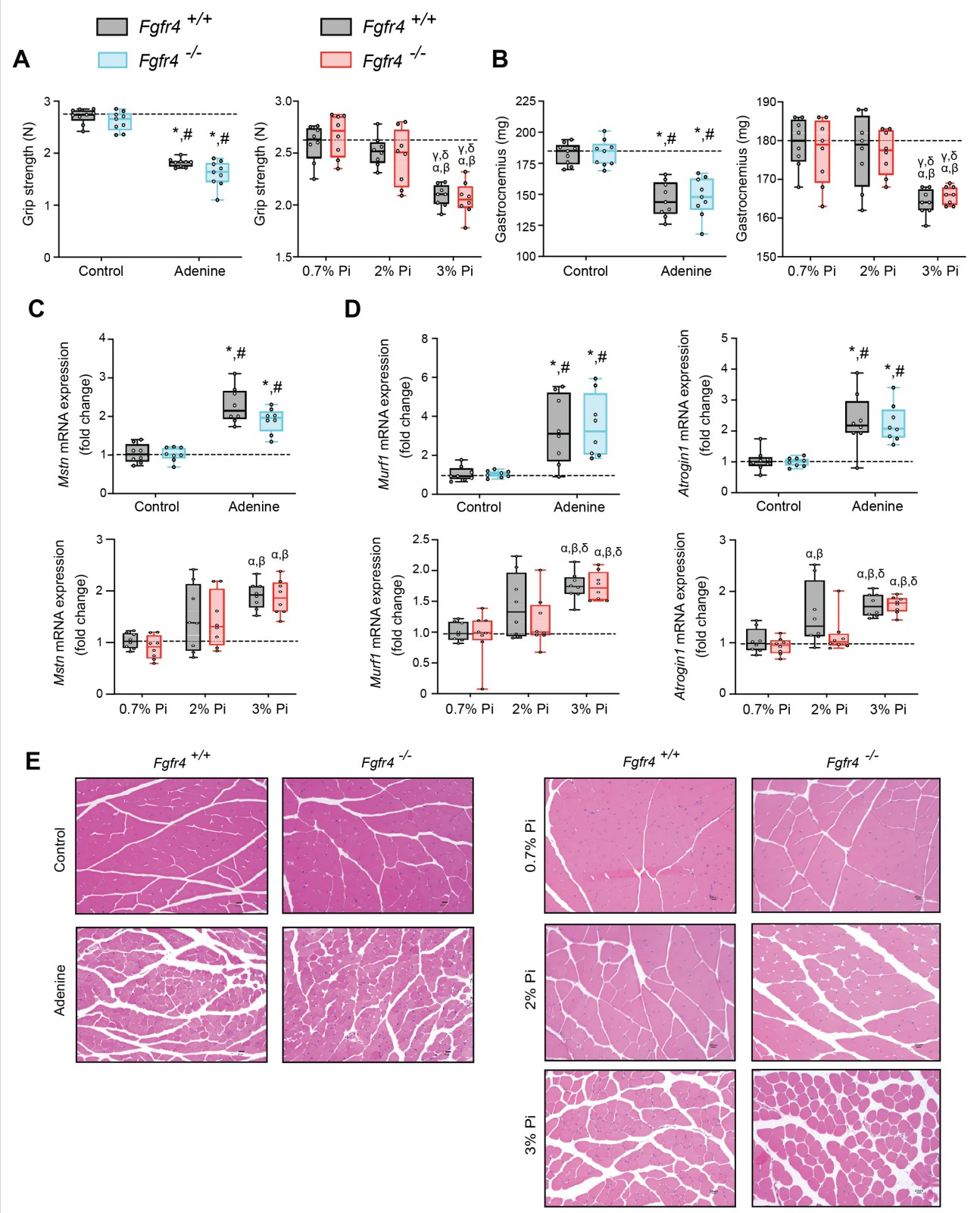

**Figure 3.** Mouse models of hyperphosphatemia exhibit signs of skeletal muscle wasting which are independent of FGF23-FGFR4 signaling. (**A**) Grip strength. (**B**) Gastrocnemius weight. Quantitative polymerase chain reaction (qPCR) analysis of *Mstn* (**C**), *Murf1* and *Atrogin1* (**D**) expression levels in gastrocnemius tissue. (**E**) Representative gross pathology of H&E-stained gastrocnemius sections (scale bar, 20 μm). All values are mean ± standard error of the mean (SEM) ((*n* = 8–9 mice/group; *p ≤ 0.05 vs. *Fgfr4*[+/+] + control diet, #p ≤ 0.05 vs. *Fgfr4*[−/−] + control diet); (*n* = 8 mice/group; αp ≤ 0.05 vs.

*Figure 3 continued on next page*

*Figure 3 continued*

$Fgfr4^{+/+}$ + 0.7% Pi diet, $^\beta$p ≤ 0.05 vs. $Fgfr4^{-/-}$ + 0.7% Pi diet, $^\gamma$p ≤ 0.05 vs. $Fgfr4^{+/+}$ + 2% Pi diet, $^\delta$p ≤ 0.05 vs. $Fgfr4^{-/-}$ + 2% Pi diet)) where statistical analyses were calculated by two-way analysis of variance (ANOVA) followed by Tukey's multiple comparison post hoc test. Dotted lines indicate median $Fgfr4^{+/+}$ + control diet or $Fgfr4^{+/+}$ + 0.7% Pi diet measurements.

The online version of this article includes the following figure supplement(s) for figure 3:

**Figure supplement 1.** Models of hyperphosphatemia exhibit signs of skeletal muscle wasting and low Pi feeding in $Col4a3^{-/-}$ (Alport syndrome) mice counteracts muscle dysfunction.

---

in $Col4a3^{-/-}$ mice by treatment, indicating increased iron mobilization and decreased iron restriction (***Figure 4—figure supplement 1C, D***). These data indicate that dietary Pi restriction improves hematological responses and alleviates hypoferremia.

As liver Pi accumulation is increased in adenine-induced CKD (***Figure 1H***), we next explored if low Pi diet treatment reduces pathologic liver Pi deposits in progressive CKD. Compared to wild-type mice on normal diet, liver Pi levels were increased in $Col4a3^{-/-}$ mice and were reduced on low Pi diet (***Figure 4H***). Taken together, our data demonstrate that Pi restriction as a dietary intervention, in a genetic model of progressive CKD, reduces pathologic Pi accumulation in the liver and alleviates the severity of renal injury and functional iron deficiency.

## Low Pi feeding counteracts signs of skeletal muscle wasting in $Col4a3-/-$ (Alport ) mice

To determine if reducing hyperphosphatemia limits skeletal muscle wasting, an important complication of CKD (***Verzola et al., 2019***), we analyzed skeletal muscle from wild-type ($Col4a3^{+/+}$) and Alport ($Col4a3^{-/-}$) mice subjected to either a normal diet (0.6% Pi) or a low Pi diet treatment (0.2% Pi) (***Figure 3***). Compared to wild-type mice on normal diet, $Col4a3^{-/-}$ mice showed significant reduction in grip strength which was improved by low Pi diet (***Figure 5A***). Gastrocnemius mass was also reduced in $Col4a3^{-/-}$ mice, and treatment tended to improve muscle weight (***Figure 5B***). In particular, gastrocnemius $Mt1$ transcript levels were significantly elevated in $Col4a3^{-/-}$ mice compared to wild-type mice on normal diet and were reduced by low Pi diet (***Figure 3—figure supplement 1C***). These results suggest that Pi restriction as a dietary intervention may improve skeletal muscle abnormalities in CKD. Furthermore, compared to wild-type mice on normal chow, $Col4a3^{-/-}$ mice displayed significant elevations in gastrocnemius $Mstn$ transcript levels which was reduced by treatment (***Figure 5C***). Additionally, $Col4a3^{-/-}$ mice on normal diet showed increased gastrocnemius $Murf1$ and $Atrogin1$ transcript levels, which were also reduced by low Pi diet (***Figure 5D***). These data support the notion that dietary Pi restriction, as a treatment in $Col4a3^{-/-}$ mice, reduces myostatin synthesis and subsequent atrophy-related gene programs.

As with adenine-induced CKD mice, or mice fed high phosphate diet, gastrocnemius tissue sections from $Col4a3^{-/-}$ mice showed smaller muscle fiber size compared to wild-type controls (***Figure 5E***). However, $Col4a3^{-/-}$ mice fed low Pi diet showed improved muscle fiber size (***Figure 5E***). Taken together, these data suggest that hyperphosphatemia affects skeletal muscle wasting in Alport mice, possibly by exacerbating systemic inflammatory cytokine concentrations and their catabolic effects on muscle.

## Pi targets hepatocytes and increases expression of inflammatory cytokines and hepcidin

Having shown that inflammation, hypoferremia, and muscle wasting induced by high Pi are independent of FGF23-FGFR4 signaling, we tested if Pi directly affects inflammatory cytokine and hepcidin expression in mouse primary hepatocytes.

We first analyzed the expression profile of the three families of Na/Pi cotransporters (types I–III). Quantitative polymerase chain reaction (qPCR) analysis detected high levels of $Slc20a1$ and $Slc20a2$ (encoding for PiT-1 and PiT-2), but not $Slc17a1$ and $Slc17a3$ (encoding for Npt-1 and Npt-4), or $Slc34a1$, $Slc34a2$ and $Slc34a3$ (encoding for NaPi-2a-c) (***Figure 6—figure supplement 1A***). This analysis indicates that type III Na/Pi cotransporters are the predominant Na/Pi family in primary hepatocyte cultures.

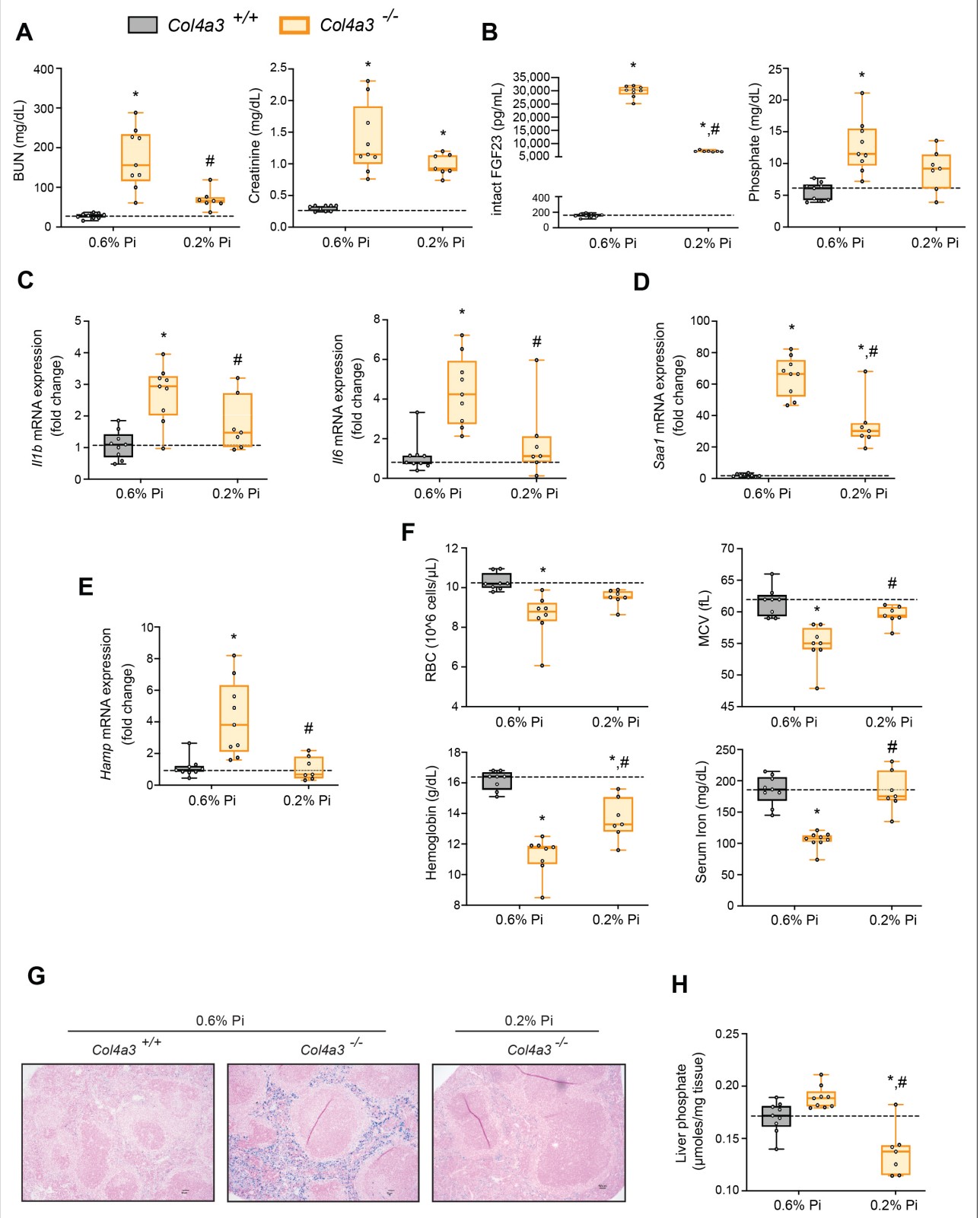

**Figure 4.** Low Pi feeding limits functional iron deficiency in *Col4a3*[−/−] (Alport ) mice. BUN, serum creatinine (**A**), serum FGF23 and serum Pi levels (**B**). Quantitative polymerase chain reaction (qPCR) analysis of *Il1b*, *Il6*, and *Saa1* (**C, D**) and *Hamp* (**E**) expression levels in liver tissue. (**F**) CBC analysis. (**G**) Representative gross pathology of Perls' Prussian blue-stained spleen sections (scale bar, 50 µm). Larger magnification is shown in supplementary figure and legends. (**H**) Liver Pi levels. All values are mean ± standard error of the mean (SEM; *n* = 7–9 mice/group; *p ≤ 0.05 vs. *Col4a3*[+/+] + 0.6% Pi diet, #p

*Figure 4 continued on next page*

*Figure 4 continued*

≤ 0.05 vs. *Col4a3*$^{-/-}$ + 0.6% Pi diet) where statistical analyses were calculated by two-way analysis of variance (ANOVA) followed by Tukey's multiple comparison post hoc test. Dotted lines indicate median *Col4a3*$^{+/+}$ + 0.6% Pi diet measurements.

The online version of this article includes the following figure supplement(s) for figure 4:

**Figure supplement 1.** Low Pi feeding limits functional iron deficiency in *Col4a3*$^{-/-}$ (Alport ) mice.

Based on studies demonstrating that high extracellular Pi activates signaling pathways such as MAPK and NFκB (*Chavkin et al., 2015*; *Zhao et al., 2011*), we assessed if MAPK, STAT3, and NFκB signaling are activated in cultured hepatocytes in response to treatments with FGF23 or graded concentrations of Pi. Treatment with TNFα or IL6 was used as a positive control for activation of

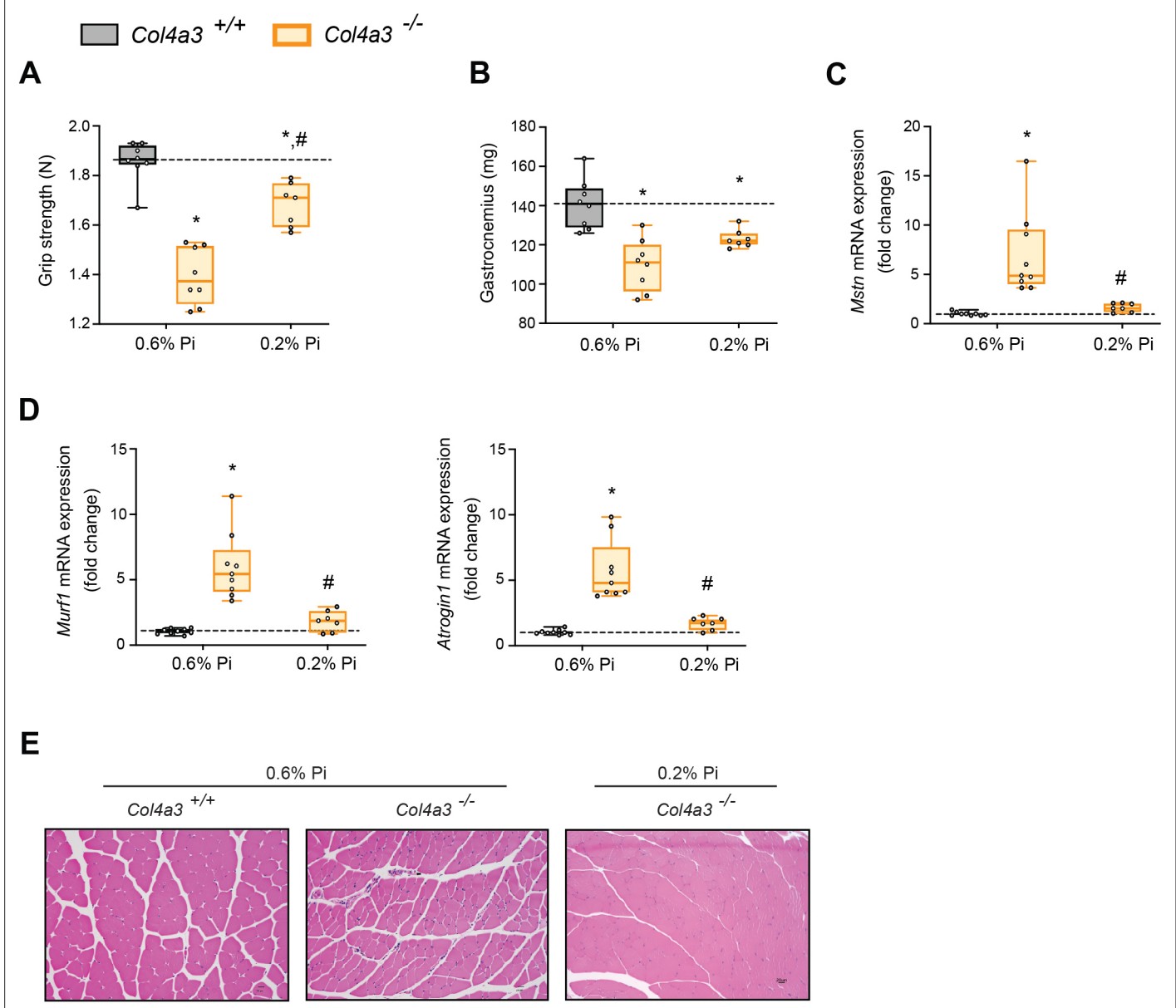

**Figure 5.** Low Pi feeding counteracts signs of skeletal muscle wasting in *Col4a3*$^{-/-}$ (Alport ) mice. (**A**) Grip strength. (**B**) Gastrocnemius weight. Quantitative polymerase chain reaction (qPCR) analysis of *Mstn* (**C**), *Murf1* and *Atrogin1* (**D**) expression levels in gastrocnemius tissue. (**E**) Representative gross pathology of H&E-stained gastrocnemius sections (scale bar, 20 μm). All values are mean ± standard error of the mean (SEM; $n$ = 7–9 mice/group; *p ≤ 0.05 vs. *Col4a3*$^{+/+}$ + 0.6% Pi diet, #p ≤ 0.05 vs. *Col4a3*$^{-/-}$ + 0.6% Pi diet) where statistical analyses were calculated by two-way analysis of variance (ANOVA) followed by Tukey's multiple comparison post hoc test. Dotted lines indicate median *Col4a3*$^{+/+}$ + 0.6% Pi diet measurements.

these established networks regulating inflammatory gene expression. Immunoblot analysis of ERK1/2, STAT3, and NFκB showed that Pi treatments increased phosphorylated NFκB levels without changing total NFκB expression (*Figure 6A*, *Figure 6—source data 1A*). Increased concentrations of $Na_2SO_4$, a salt generating another anionic species, had no effect on phospho-NFκB levels, indicating this response was specific to elevated Pi and not an unspecific response to increased anions. Pi treatment did not affect pERK1/2 or pSTAT3, and FGF23 had no effect on any of the pathways examined.

We next analyzed gene expression by qPCR of inflammatory cytokines and acute phase proteins in isolated hepatocytes treated with increasing concentrations of Pi or $Na_2SO_4$ with LPS and IL6 treatments used as positive control. Elevations in *Il1b*, *Il6*, and *Saa1* transcript levels were noted not only following LPS and IL6 but also Pi treatments (*Figure 6B, C*). Treatments with $Na_2SO_4$ had no effect on gene expression. As inflammation is a known mediator of hepcidin synthesis, we analyzed *Hamp* mRNA levels. LPS, IL6, and Pi treatments all elevated *Hamp* transcript levels when compared to control (*Figure 6D*). These data indicate that high extracellular Pi can act on hepatocytes to increase the synthesis of inflammatory cytokines and hepcidin.

Given that inflammation and NFκB signaling regulate PiT-1 expression (*Koumakis et al., 2019*) and primary hepatocytes express *Slc20a1* and *Slc20a2* (*Figure 6—figure supplement 1A*), we examined if *Slc20a1* and/or *Slc20a2* mRNA levels were altered following LPS, IL6, or Pi treatment. Expression analysis showed that Pi significantly increased *Slc20a1* transcript levels in a dose-dependent manner but had no effect on *Slc20a2* expression (*Figure 6E*, *Figure 6—figure supplement 1B*). To determine if this result is an action of hepatocytes sensing high extracellular Pi, we cotreated hepatocytes with phosphonoformic acid (PFA), a compound which is reported to inhibit chemisorption of calcium–Pi clusters, as aggregate formation is a byproduct of increased extracellular Pi (*Villa-Bellosta et al., 2007*; *Villa-Bellosta and Sorribas, 2009*). In the presence of PFA, Pi-induced *Slc20a1* expression was reduced compared to vehicle-treated control cells (*Figure 6F*). Interestingly, PFA also altered the effects of LPS and IL6 on *Slc20a1* expression. To confirm whether inhibiting high extracellular Pi and/or aggregate byproducts disrupts downstream actions of increased extracellular Pi, we cotreated hepatocytes with Pi and PFA, and observed that PFA interfered with Pi-induced effects on phospho-NFκB levels without changing total NFκB expression (*Figure 6G*, *Figure 6—source data 1G*).

Furthermore, when isolated hepatocytes were treated with either LPS, IL6, or Pi in the presence or absence of PFA, the significant elevations in *Il1b*, *Il6*, *Saa1*, and *Hamp* transcript levels following Pi treatments were reversed in the presence of PFA (*Figure 6H–J*). Interestingly, PFA slightly altered the effects of LPS on *Il6* expression and IL6 on *Hamp* expression (*Figure 6H–J*). Taken together, our results indicate that in primary hepatocyte cultures type III Na/Pi cotransporters are predominant, and that increased extracellular Pi activates NFκB signaling, increases PiT-1 expression and induces inflammatory cytokine and hepcidin production.

## Pi induces hepcidin expression via paracrine IL1β and IL6 signaling

Next, we explored if NFκB is a necessary mediator for inflammatory cytokine and hepcidin regulation by high extracellular Pi in vitro. We treated mouse primary hepatocytes with either LPS or graded Pi concentrations in the presence or absence of a selective NFκB pharmacologic inhibitor, BAY 11-7082 (*Koumakis et al., 2019*; *Pierce et al., 1997*). Significant elevations in *Il1b*, *Il6*, and *Hamp* mRNA levels were detected following LPS or Pi treatments, and these were attenuated by BAY 11-7082 (*Figure 7A, B*). Similar effects of BAY 11-7082 on *Saa1*, *haptoglobin (Hp)* and *Slc20a1* mRNA expression were also observed, corroborating reports that NFκB directly regulates PiT-1 abundance (*Figure 7—figure supplement 1A–C*).

As inflammation directly regulates hepcidin, we explored if Pi-induced hepcidin expression is a response resulting from direct or indirect actions of NFκB. Testing indirect effects, we cotreated primary hepatocytes with either LPS or graded Pi concentrations in the presence or absence of anti-IL1β antibody, anti-IL6 antibody, or both neutralizing antibodies in combination. Expression analysis detected significant elevations in *Hamp* mRNA levels following LPS or Pi treatments and was blunted by addition of either antibody alone or in combination (*Figure 7C*). To ensure treatments generated endogenous IL1β and IL6 protein, we analyzed *Saa1* and *Hp* mRNA expression, as both genes are regulated by IL1β and IL6 (*Zhou et al., 2016*). Compared to vehicle-treated control cells, *Saa1* and *Hp* transcript levels were reduced in the presence of either antibody alone or in combination, following LPS or Pi treatment (*Figure 7—figure supplement 1D, E*). Collectively, these results show Pi-mediated

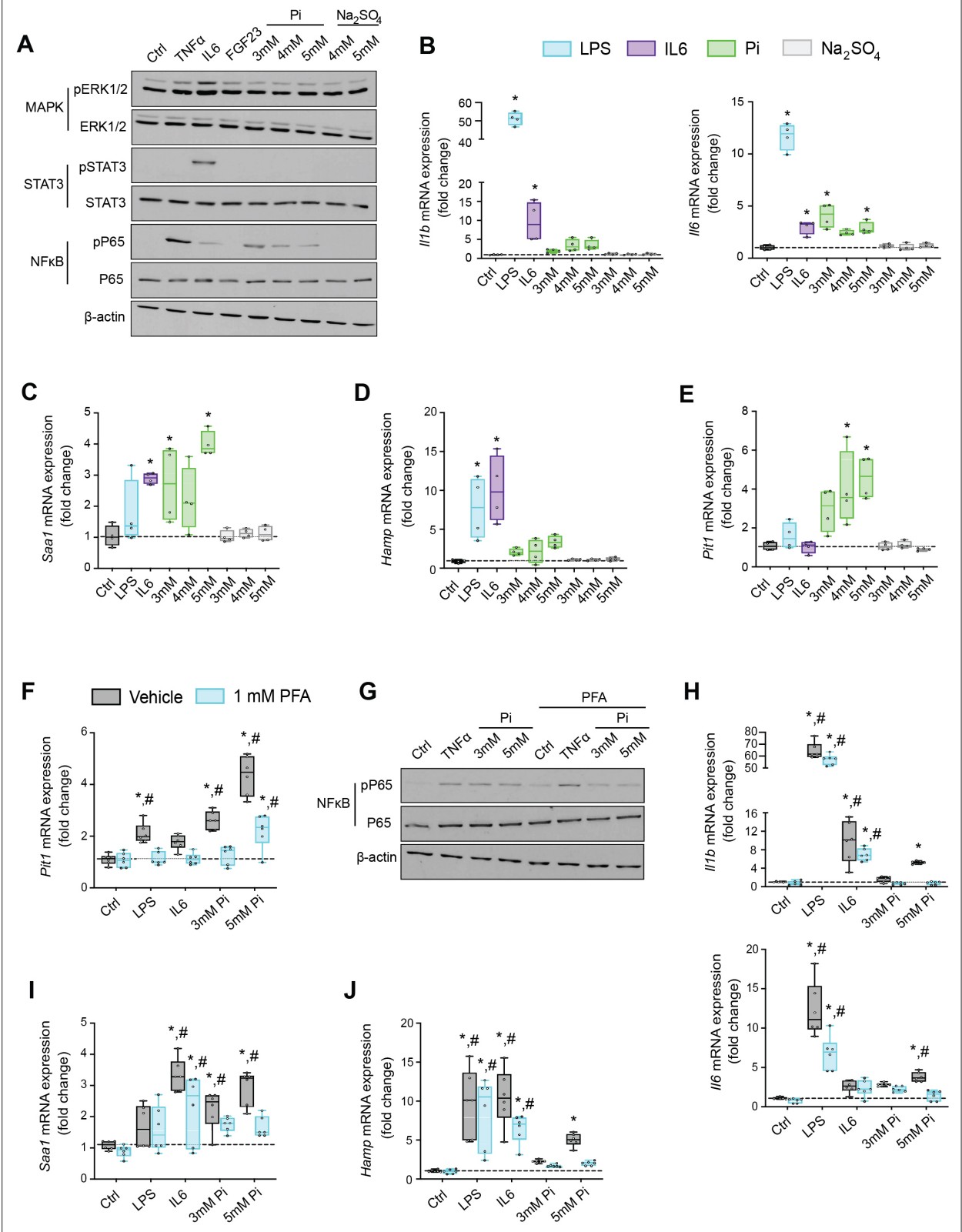

**Figure 6.** Pi targets hepatocytes and increases expression of inflammatory cytokines and hepcidin. (**A**) Immunoblot analysis of total protein extracts from primary hepatocytes (*n* = 5 independent isolations). β-Actin serves as loading control. Quantitative polymerase chain reaction (qPCR) analysis of *Il1b*, *Il6*, *Saa1* (**B, C**), *Hamp* (**D**), and *Slc20a1* (**E**) expression levels in primary hepatocytes; values are mean ± standard error of the mean (SEM; *n* = 4 independent isolations; *p ≤ 0.05 vs. control [Ctrl]). Dotted lines indicate median Ctrl measurements. (**F**) qPCR analysis of *Slc20a1* expression levels in primary

*Figure 6 continued on next page*

Figure 6 continued

hepatocytes following stimuli, with or without phosphonoformic acid (PFA); values are mean ± standard error of the mean (SEM; $n = 6$ independent isolations; *p ≤ 0.05 vs. vehicle control [Ctrl], #p ≤ 0.05 vs. 1 mM PFA Ctrl). Dotted lines indicate median vehicle Ctrl measurements. (**G**) Immunoblot analysis of total and phosphorylated p65 (NF $\kappa$ B) protein levels from primary hepatocytes following stimuli, with or without PFA ($n = 5$ independent isolations). β-Actin serves as loading control. (**H–J**) qPCR analysis of *Il1b*, *Il6*, *Saa1* (**H–I**) and *Hamp* (**J**) expression levels in primary hepatocytes following stimuli, with or without PFA; values are mean ± standard error of the mean (SEM; $n = 6$ independent isolations; *p ≤ 0.05 vs. vehicle control [Ctrl], #p ≤ 0.05 vs. 1 mM PFA Ctrl) where statistical analyses were calculated by one-way analysis of variance (ANOVA; **B–E**) or by two-way ANOVA (**F, H–J**) followed by Tukey's multiple comparison post hoc test. Dotted lines indicate median vehicle Ctrl measurements.

The online version of this article includes the following source data and figure supplement(s) for figure 6:

**Source data 1.** Original western blots.

**Figure supplement 1.** Pi targets hepatocytes and increases expression of inflammatory cytokines and hepcidin.

hepcidin production in cultured hepatocytes is a result of NFκB amplifying PiT-1 expression, which in turn, might intensify high extracellular Pi to augment NFκB regulated inflammatory gene programs, prompting the induction of required cytokines IL1β and IL6 to mediate hepcidin production.

## Discussion

We report that hyperphosphatemia, either as a result of adenine-induced CKD or dietary Pi excess, increases inflammation to exacerbate anemia and skeletal muscle wasting (**Figure 8**). These complications are associated with increased liver Pi levels, which correlated with serum Pi concentrations. Supplying a low Pi diet treatment to Alport mice, a genetic model of CKD, results in beneficial outcomes that reduce functional iron deficiency and skeletal muscle wasting. Furthermore, our mechanistic in vitro studies indicate that Pi elevations induce hepatic production of IL6 and IL1β to increase hepcidin expression in hepatocytes, a potential causative link between hyperphosphatemia, anemia, and skeletal muscle dysfunction.

Previously, we reported pathologic FGF23-FGFR4 signaling might contribute to excess inflammatory mediators (**Singh et al., 2016**), and we now followed up on the FGF23 inflammatory role in clinically relevant CKD models in vivo. Here, we examined wild-type (*Fgfr4*[+/+]) and constitutive FGFR4 knockout (*Fgfr4*[−/−]) mice subjected to adenine diet or a graded dietary Pi load. We found that on adenine, both *Fgfr4*[+/+] and *Fgfr4*[−/−] mice show comparable macroscopic parameters (**Table 1**) and degrees of functional iron deficiency (**Figure 1**). These findings coincide with greater levels of liver Pi, which raises the possibility that pathologic Pi deposits, in tissues apart from the vasculature, may contribute to additional complications in CKD (**Komaba and Fukagawa, 2016**).

Clinical reports indicate CKD patients have dysregulated Pi handling (**Chang et al., 2014**; **Isakova et al., 2009**) and together with the consumption of foods and drinks rich in Pi-based additives, such as in a Westernized diet, extrarenal Pi accumulation may occur (**Isakova et al., 2008**). A recent study utilizing animal models supports this postulate, demonstrating that excess Pi leads to depositions into tissues such as the vasculature (**Zelt et al., 2019**). Moreover, a recent report indicates the major source of body Pi removed during hemodialysis in CKD patients, is from cells releasing intracellular Pi (**Chazot et al., 2021**). As the serum Pi compartment represents a small fraction of total body Pi, and the uptake of excess Pi by tissues is recognized as a detrimental trigger, it is important to examine the degree of pathologic Pi accumulation in nonvascular tissue and whether it exacerbates complications in CKD, such as anemia and muscle wasting.

We employed a graded Pi diet to study the effects of excess Pi on the liver (**Figure 2**). Studies show conflicting results toward renal and liver health, following supplementation of a high Pi diet (**Baquerizo et al., 2003**; **Haut et al., 1980**; **Ugrica et al., 2021**). In our study, no significant changes in macroscopic parameters were observed following dietary Pi overload (**Table 2**). Also, no pathologic changes in kidney (**Figure 2—figure supplement 1**) or liver were detected (**Figure 2—figure supplement 2**). However, mice on a 3% Pi diet exhibit increased liver inflammation and *Hamp* expression, which corroborates previous observation that high dietary Pi influences hepcidin production (**Nakao et al., 2015**). These results coincide with positive correlations between liver Pi and liver *Hamp* mRNA expression, with onset of this correlation preceding significant elevations in serum Pi. Despite these data suggesting that liver Pi influences liver hepcidin production, our finding might indicate that increased extrarenal Pi accumulation provides a reservoir for storing excess Pi until tissue accumulation

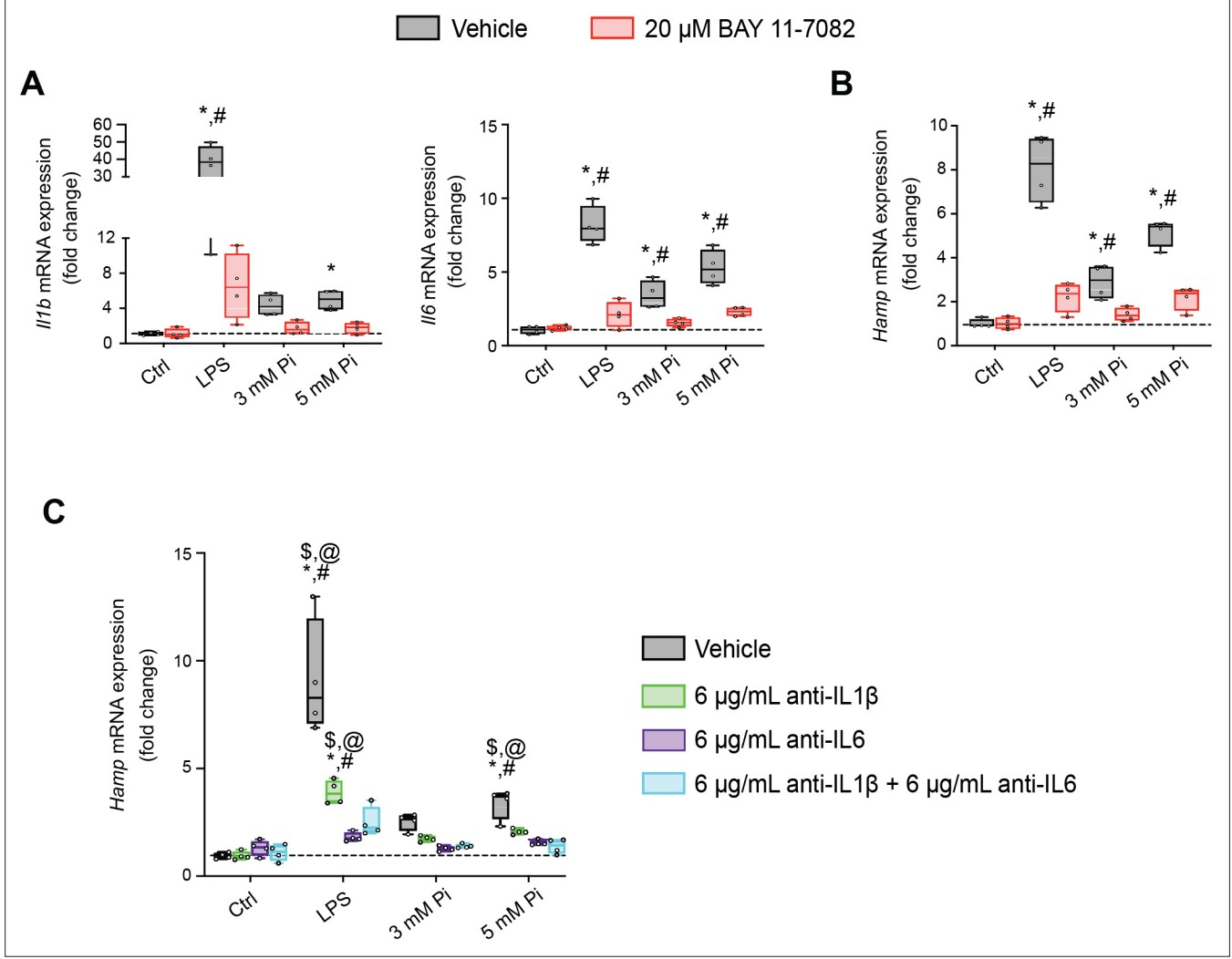

**Figure 7.** Pi induces hepcidin expression via paracrine IL1β and IL6 signaling. Quantitative polymerase chain reaction (qPCR) analysis of *Il1b*, *Il6* (**A**) and *Hamp* (**B**) expression levels in primary hepatocytes following stimuli, with or without BAY 11-7082; values are mean ± standard error of the mean (SEM; *n* = 4 independent isolations; *p ≤ 0.05 vs. vehicle control [Ctrl], #p ≤ 0.05 vs. 20 µM BAY 11-7082 Ctrl). Dotted lines indicate median vehicle Ctrl measurements. (**C**) qPCR analysis of *Hamp* expression levels in primary hepatocytes following stimuli with or without anti-IL1β, anti-IL6, or both antibodies in combination; values are mean ± standard error of the mean (SEM; *n* = 4 independent isolations; *p ≤ 0.05 vs. vehicle control [Ctrl], #p ≤ 0.05 vs. anti-IL1β Ctrl, $p ≤ 0.05 vs. anti-IL6 Ctrl, @p ≤ 0.05 vs. anti-IL1β + anti-IL6 Ctrl) where statistical analyses were calculated by two-way analysis of variance (ANOVA) followed by Tukey's multiple comparison post hoc test. Dotted lines indicate median vehicle Ctrl measurements.

The online version of this article includes the following figure supplement(s) for figure 7:

**Figure supplement 1.** Pi induces hepcidin expression via paracrine IL1β and IL6 signaling.

achieves saturation, in which case the serum Pi compartment then gradually rises, resulting in hyperphosphatemia. Furthermore, our data suggest that prolonged exposure to Pi, if not maintained in adequate quantities, might trigger pathologic outcomes, as mice on 3% Pi show a noticeable degree of hypoferremia. None of the observed effects of the high Pi diet were mediated by FGFR4, as *Fgfr4*[−/−] mice were comparable to wild-type mice in all the parameters measured. However, this work cannot exclude the potential of alternative FGFR isoforms which might mediate the effects of excess FGF23 toward functional iron deficiency following either adenine or 3% Pi diet, as a previous report exhibits the utilization of a single intraperitoneal injection of FGF23 blocking peptide was sufficient to rescue anemia (*Agoro et al., 2018*).

Excess dietary Pi was shown to directly exacerbate intestinal inflammation in a model of experimental colitis (*Sugihara et al., 2017*) and that reducing dietary Pi provides beneficial outcomes toward systemic inflammation, accelerated aging, and survival, as demonstrated in a model of senescence

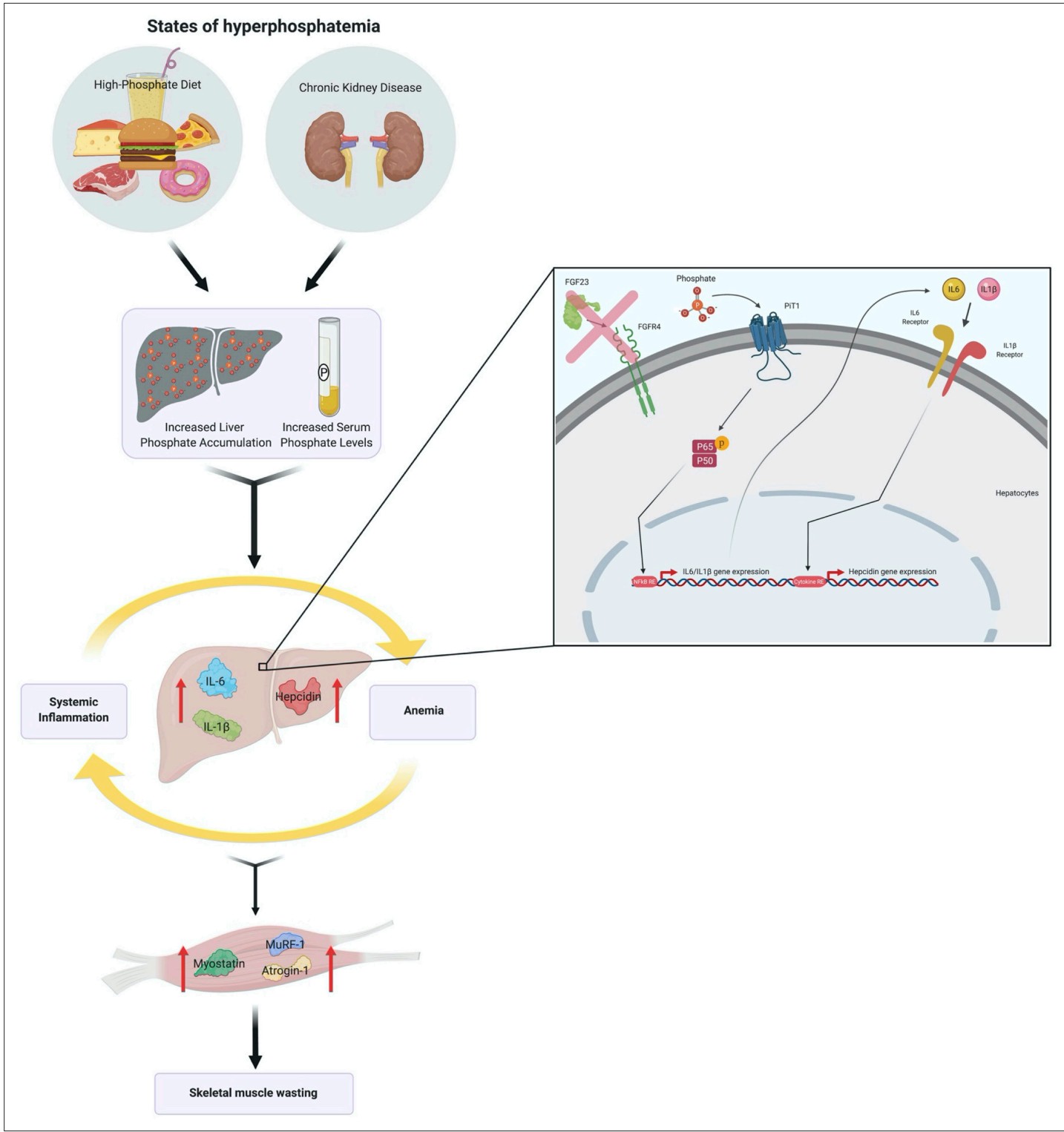

**Figure 8.** Schematic of the effects of hyperphosphatemia on systemic inflammation, hypoferremia, and skeletal muscle wasting.

(*Morishita et al., 2001*). We likewise observed increased inflammation in our hyperphosphatemic mouse models. Inflammatory cytokines can directly target skeletal muscle cells to induce muscle wasting by increasing myostatin production (*Zhang et al., 2013*), which both together, enhances protein degradation and reduces protein synthesis, as CKD illustrates catabolic conditions which are attributable to the vicious cycle generated between mineral dyshomeostasis and inflammation (*Czaya*

**Table 1.** Macroscopic parameters of *Fgfr4*$^{+/+}$ and *Fgfr4*$^{-/-}$ mice receiving control and adenine diet.

| | *Fgfr4*$^{+/+}$ + control diet | *Fgfr4*$^{-/-}$ + control diet | *Fgfr4*$^{+/+}$ + adenine diet | *Fgfr4*$^{-/-}$ + adenine diet |
|---|---|---|---|---|
| Body weight (g) | 30.1 ± 0.9 | 30.2 ± 0.3 | 16.8*# ± 0.5 | 17.5*# ± 0.4 |
| Liver weight (g) | 1.14 ± 0.05 | 1.22 ± 0.08 | 0.74*# ± 0.03 | 0.78*# ± 0.05 |
| Spleen weight (mg) | 75.0 ± 2.2 | 76.0 ± 1.7 | 53.3*# ± 3.3 | 56.0*# ± 3.2 |
| Left kidney weight (mg) | 181.8 ± 8.5 | 173.2 ± 8.3 | 122.2*# ± 4.9 | 101.8*# ± 7.2 |
| Right kidney weight (mg) | 184.7 ± 10.7 | 175.2 ± 8.2 | 124.2*# ± 4.1 | 102.8*# ± 7.7 |

Values are expressed as mean ± standard error of the mean (SEM). Comparison between groups was performed in form of a two-way analysis of variance (ANOVA) followed by a post hoc Tukey test. A level of $p < 0.05$ was accepted as statistically significant; $N = 9$/group; *$p \leq 0.05$ vs. *Fgfr4*+/+ + control diet, #$p \leq 0.05$ vs. *Fgfr4*−/− + control diet.

*and Faul, 2019b*). We indeed observed skeletal muscle wasting in adenine-induced CKD, high phosphate diet, and genetic model of CKD. Ablation of FGFR4 in mice did not improve skeletal muscle function following adenine or 3% Pi diets, suggesting hyperphosphatemia rather than pathologic FGF23-FGFR4 signaling might be the cause of skeletal muscle abnormalities (*Figure 3*). This hypothesis is supported by reports demonstrating excess Pi influences skeletal muscle dysfunction (*Acevedo et al., 2016*; *Chen et al., 2018*; *Chung et al., 2020*), although it is possible additional FGFR isoforms directly promote skeletal muscle wasting due to excess FGF23 following adenine or high Pi diet. However, a recent report suggests that FGF23 does not directly affect skeletal muscle dysfunction (*Avin et al., 2018*).

To assess whether reducing hyperphosphatemia can improve inflammation, anemia, and skeletal muscle wasting, we exposed Alport mice, a genetic model of progressive CKD, to a low Pi diet treatment. Indeed, despite severe elevations in serum FGF23, dietary Pi restriction limited functional iron deficiency (*Figure 4*). Our data also show liver Pi levels were reduced in Alport mice following low Pi diet treatment, in comparison to Alport mice on normal diet. These findings provide strong evidence that hyperphosphatemia, specifically pathologic liver Pi accumulation, rather than pathologic FGF23-FGFR4 signaling, might exacerbate inflammation and hypoferremia. Skeletal muscle function and mass were also improved by a low Pi diet in Alport mice, along with decreased expression of muscle myostatin and atrophy-related gene programs, culminating in larger myofiber size. These findings suggest the contribution of hyperphosphatemia to skeletal muscle wasting may result from an indirect mechanism that regulates inflammatory cytokines and their pleotropic activities, such as increased liver-derived IL1β and IL6, which might increase overall systemic levels that effectively target skeletal muscle. Despite this postulate, further work will be needed to determine if high extracellular Pi directly targets skeletal muscle cells to affect muscle function. Nonetheless, these data add to the growing list of adverse outcomes of Pi toxicity such as gingivitis, accelerated aging, vascular calcification and tumorigenesis (*Erem and Razzaque, 2018*). Furthermore, Alport mice on low Pi diet treatment displayed a reduced degree of pathologic kidney function, alterations in kidney morphology, and macroscopic parameters (*Figure 4*, *Figure 4—figure supplement 1*, *Table 3*). Thus, we cannot exclude that these beneficial outcomes in Alport mice observed on treatment may be a repercussion of slightly improved kidney function, as a recent report demonstrates elevated Pi concentrations directly affect proximal tubular function (*Shiizaki et al., 2021*).

**Table 2.** Macroscopic parameters of *Fgfr4*$^{+/+}$ and *Fgfr4*$^{-/-}$ mice receiving a graded dietary Pi load.

| | *Fgfr4*$^{+/+}$ + 0.7% Pi diet | *Fgfr4*$^{-/-}$ + 0.7% Pi diet | *Fgfr4*$^{+/+}$ + 2% Pi diet | *Fgfr4*$^{-/-}$ + 2% Pi diet | *Fgfr4*$^{+/+}$ + 3% Pi diet | *Fgfr4*$^{-/-}$ + 3% Pi diet |
|---|---|---|---|---|---|---|
| Body weight (g) | 32.0 ± 1.0 | 31.9 ± 1.0 | 30.3 ± 0.9 | 31.6 ± 0.9 | 29.5 ± 0.3 | 29.5 ± 0.4 |
| Liver weight (g) | 1.17 ± 0.04 | 1.15 ± 0.05 | 1.24 ± 0.03 | 1.20 ± 0.05 | 1.23 ± 0.5 | 1.19 ± 0.4 |
| Spleen weight (mg) | 76.3 ± 2.4 | 68.3 ± 1.7 | 77.0 ± 2.8 | 76.6 ± 2.8 | 76.6 ± 2.1 | 74.0 ± 1.7 |
| Left kidney weight (mg) | 146.5 ± 4.7 | 143.6 ± 4.5 | 156.0 ± 2.9 | 152.6 ± 3.4 | 153.4 ± 4.1 | 152.9 ± 3.7 |
| Right kidney weight (mg) | 149.0 ± 3.8 | 151.5 ± 3.7 | 152.9 ± 3.2 | 145.6 ± 3.5 | 152.5 ± 2.5 | 149.6 ± 3.9 |

Values are expressed as mean ± standard error of the mean (SEM). Comparison between groups was performed in form of a two-way analysis of variance (ANOVA) followed by a post hoc Tukey test. No level of statistical significance was accepted between groups; $N = 8$/group.

**Table 3.** Macroscopic parameters of Alport mice receiving either a 0.6% Pi diet or 0.2% Pi diet.

| | Col4a3[+/+] + 0.6% Pi diet | Col4a3[–/–] + 0.6% Pi diet | Col4a3[–/–] + 0.2% Pi diet |
|---|---|---|---|
| Body weight (g) | 26.3 ± 0.6 | 16.3* ± 0.6 | 22.2*[#] ± 0.6 |
| Liver weight (g) | 1.03 ± 0.04 | 0.68* ± 0.03 | 0.90[#] ± 0.03 |
| Spleen weight (mg) | 72.2 ± 2.3 | 56.6* ± 2.4 | 65.9 ± 2.1 |
| Left kidney weight (mg) | 145.3 ± 1.6 | 124.1* ± 2.6 | 130.1* ± 2.1 |
| Right kidney weight (mg) | 147.4 ± 1.6 | 123.0* ± 3.4 | 133.6* ± 3.1 |

Values are expressed as mean ± standard error of the mean (SEM). Comparison between groups was performed in form of a two-way analysis of variance (ANOVA) followed by a post hoc Tukey test. A level of $p < 0.05$ was accepted as statistically significant; $N = 7–9$/group; *$p ≤ 0.05$ vs. Col4a3+/+ + 0.6% Pi diet, [#]$p ≤ 0.05$ vs. Col4a3–/– + 0.6% Pi diet.

Importantly, we identify a molecular mechanism that potentially links hyperphosphatemia to anemia and skeletal muscle dysfunction. Utilizing mouse primary hepatocytes, we demonstrate high extracellular Pi activates NFκB signaling and leads to subsequent inflammatory cytokine and hepcidin production (*Figure 6*). Employing PFA, a compound reported to reduce calcium–Pi deposition and cluster formation, as aggregates are a byproduct of increased extracellular Pi (*Villa-Bellosta et al., 2007*; *Villa-Bellosta and Sorribas, 2009*), we confirm NFκB activation is a direct action of Pi targeting hepatocytes, which prompts subsequent *Slc20a1* mRNA expression, as observed from our BAY 11-7082 findings. This high extracellular Pi-NFκB signaling axis is observed in other reservoirs such as vascular smooth muscle cells and ex vivo kidney slices, as well as adjuncts the rewiring of various signaling networks that control cellular homeostasis following excessive Pi exposure (*He et al., 2021*; *Rodríguez-Ortiz et al., 2020*; *Voelkl et al., 2018*; *Zhao et al., 2011*). Although this reaffirms NFκB directly influences PiT-1 levels (*Koumakis et al., 2019*), it does not identify if our observations are dependent or independent of Pi translocation, as extracellular Pi might associate with various PiT-1 extracellular regions to influence PiT-1/PiT-1 homodimerization, PiT-1/PiT-2 heterodimerization or a conformational change in PiT-1 to initiate the activation of selected binding partners which mediate downstream signaling events (*Bon et al., 2018*; *Forand et al., 2016*). In addition to the amplified hepatic PiT-1 abundance and recognition of NFκB as a necessary mediator of high extracellular Pi in hepatocytes, we show that the effect of Pi on hepcidin requires the indirect actions of NFκB and biological activities of endogenous IL1β and IL6 proteins secreted by hepatocytes, as elucidated by our cell-based neutralization assay of these targeted cytokines (*Figure 7*). Based on these findings, we speculate increased liver Pi deposits might underlie a clinical association between elevated body Pi and inflammation, where the prolonged duration of tissue accumulation permits Pi in the liver to directly target hepatocytes to induce inflammatory gene programs and hepcidin expression, contributing to hypoferremia. This could explain associations of inflammation and anemia in FTC patients, in addition to patients with and without CKD, before they exhibit hyperphosphatemia (*Ramnitz et al., 2016*; *Tran et al., 2016*; *Wojcicki, 2013*).

Our study has some limitations. Although we confirm hyperphosphatemia affects specific complications, our study does not specifically address the actions of certain aggregate byproducts formed by increased extracellular Pi. Nonetheless, this principal emphasis on elevations in plasma Pi concentrations will ultimately impact the formation of byproducts. Notably, the identification of the specific Pi sensor which mediates our observed hepatic Pi actions are not definite and remains to be defined with our ongoing studies. Moreover, our study does not address the specific actions of hyperphosphatemia on bone metabolism. As bone is a reservoir of extracellular Pi, potential alterations in bone health could relay a crosstalk between bone, liver, and/or skeletal muscle, which might contribute to our reported observations.

In summary, we investigated whether hyperphosphatemia and/or pathologic FGF23-FGFR4 signaling aggravates inflammation, anemia, and skeletal muscle wasting. We establish hyperphosphatemia, as found in dietary Pi overload or in CKD, is a detrimental trigger which activates hepatic NFκB signaling to stimulate an inflammatory response, which in turn, exacerbates hypoferremia and widespread complications such as skeletal muscle wasting. Notably, these findings are independent of pathologic FGF23-FGFR4 signaling. Clinical studies have demonstrated conflicting outcomes with

traditional Pi binders in individuals with nondialysis-dependent CKD, but modern Pi binders, such as ferric citrate, demonstrate greater efficacy (*Francis et al., 2019*; *Toussaint et al., 2020*), and are being evaluated for their effect on CKD comorbidities. Furthermore, reports assessing dietary Pi restriction in animal models are scarce. Our current experimental data suggest hyperphosphatemia, in itself, is pathologic and demands further attention for alternative strategies to resolve current ineffective approaches. Treatments, such as pharmacologic inhibition of type II Na/Pi cotransporters, hold potential for therapeutic actions (*Clerin et al., 2020*; *Thomas et al., 2019*) but do not aim at altered mineral metabolism. Using in vitro studies and complementary animal models, we provide insights regarding the interconnection between altered mineral metabolism and common complications. Dietary Pi restriction might alleviate these sequelae, if sustained effectively as a clinical treatment. By elucidating direct inflammatory actions of high extracellular Pi on hepatocytes, we expose additional adverse outcomes of hyperphosphatemia, besides vascular calcification. These findings may yield new targets for therapeutic development, with emphasis on hepatic Pi actions. Moreover, as studies indicate anti-FGFR4 therapy may be beneficial toward cardiomyopathy (*Grabner et al., 2015*), our data suggest these same beneficial outcomes for inflammation, anemia, and skeletal muscle wasting would not apply. Altogether, our study features a possibility to improve CKD patient survival and specific rare genetic disorders, such as FTC, by limiting excess body Pi.

# Materials and methods

**Key resources table**

| Reagent type (species) or resource | Designation | Source or reference | Identifiers | Additional information |
|---|---|---|---|---|
| Strain, strain background (*Mus musculus*; 129/SvJ) | *Col4a3$^{tm1Dec}$* | Jackson Laboratory | Stock No. 000691 | Referred to as *Col4a3$^{-/-}$* |
| Strain, strain background (*Mus musculus*; C57Bl/6) | Global FGFR4 knockout | Gift from Dr. Chu-Xia Deng, NIDDK, Bethesda, USA | *Weinstein et al., 1998*, 125, 3615-23 | Referred to as *Fgfr4$^{-/-}$* |
| Antibody | IL6 (rat monoclonal) | R&D Systems | MP5-20F3 | For cell culture treatment (6 µg/ml) |
| Antibody | IL1β (goat polyclonal) | R&D Systems | AF-401-NA | For cell culture treatment (6 µg/ml) |
| Antibody | ERK1/2 (rabbit monoclonal) | Cell Signaling | 4695 | For WB (1:1000) |
| Antibody | STAT3 (rabbit monoclonal) | Cell Signaling | 4904 | For WB (1:1000) |
| Antibody | NFkB (rabbit monoclonal) | Cell Signaling | 8242 | For WB (1:1000) |
| Antibody | β-Actin (rabbit monoclonal) | Cell Signaling | 4970 | For WB (1:1000) |
| Antibody | Phosphorylated ERK1/2 (rabbit polyclonal) | Cell Signaling | 9101 | For WB (1:1000) |
| Antibody | Phosphorylated STAT3 (rabbit monoclonal) | Cell Signaling | 9145 | For WB (1:1000) |
| Antibody | Phosphorylated NF$\kappa$B (rabbit monoclonal) | Cell Signaling | 3033 | For WB (1:1000) |
| Antibody | Mouse IgG, HRP conjugate (goat monoclonal) | Promega | W4021 | For WB (1:2500) |
| Antibody | Rabbit IgG, HRP conjugate (goat monoclonal) | Promega | W4011 | For WB (1:2500) |
| Peptide, recombinant protein | FGF23; mouse | R&D Systems | 2629-FG | For cell culture treatment (25 ng/ml) |
| Peptide, recombinant protein | TNFα; mouse | R&D Systems | 410-MT | For cell culture treatment (100 ng/ml) |
| Peptide, recombinant protein | IL6; mouse | R&D Systems | 406 ML | For cell culture treatment (50 ng/ml) |
| Commercial assay or kit | Colorimetric Pi assay | Abcam | Ab65622 | |

*Continued on next page*

*Continued*

| Reagent type (species) or resource | Designation | Source or reference | Identifiers | Additional information |
|---|---|---|---|---|
| Commercial assay or kit | Colorimetric iron assay | Sekisui | 157-30 | |
| Commercial assay or kit | Intact FGF23 ELISA (mouse) | Quidel | 60-6800 | |
| Commercial assay or kit | Percoll gradient solution | Sigma-Aldrich | P1644 | |
| Commercial assay or kit | RNeasy Plus Mini kit | Qiagen | 74,136 | |
| Commercial assay or kit | RNeasy Plus Universal Mini kit | Qiagen | 73,404 | |
| Commercial assay or kit | iScript Reverse Transcriptase Supermix | BioRad | 1708840 | |
| Commercial assay or kit | SSoAdvanced Universal SYBR Green Supermix | BioRad | 172-5272 | |
| Commercial assay or kit | Pierce BCA Protein Assay | Thermo Fisher | 23,225 | |
| Chemical compound, drug | Lipopolysaccharide (LPS) from *E. coli* serotype 0111:B4; endotoxin agent | Invivogen | Tlrl-3pelps | For cell culture treatment (100 ng/ml) |
| Chemical compound, drug | Sodium phosphate dibasic anhydrous | Fisher | BP332-500 | |
| Chemical compound, drug | Sodium phosphate monobasic anhydrous | Fisher | BP329-1 | |
| Chemical compound, drug | Sodium Sulfate | Sigma-Aldrich | 239,313 | |
| Chemical compound, drug | Phosphonoformic acid (PFA) | Sigma-Aldrich | P6801 | |
| Chemical compound, drug | BAY 11-7082 | Selleckhem | S2913 | For cell culture treatment (20 µM) |
| Software, algorithm | GraphPad Prism | GraphPad | | |
| Other | 0.2% adenine diet | Envigo | TD.140290 | Diet for mice |
| Other | 0.15% adenine diet | Envigo | TD.170304 | Diet for mice |
| Other | Adenine control diet | Envigo | TD.170303 | Diet for mice |
| Other | 0.7% Pi diet | Envigo | TD.180287 | Diet for mice |
| Other | 2% Pi diet | Envigo | TD.08020 | Diet for mice |
| Other | 3% Pi diet | Envigo | TD.180286 | Diet for mice |
| Other | 0.6% Pi diet | Envigo | TD.200407 | Diet for mice |
| Other | 0.2% Pi diet | Envigo | TD.200406 | Diet for mice |

## Materials

Recombinant proteins used are mouse FGF23 (2629-FG, R&D Systems), mouse TNFα (410-MT, R&D Systems), and mouse IL6 (406 ML, R&D Systems). This FGF23 peptide contains an arginine to glutamine amino acid substitution at position 179 which yields it resistant to furin protease-mediated degradation, thus prolonging its half-life. Lipopolysaccharide (LPS) from *E. coli* serotype 0111:B4 (tlrl-3pelps, Invivogen) was used as endotoxin. Sodium phosphate dibasic anhydrous ($Na_2HPO_4$) (BP332-500, Fisher Scientific) and sodium phosphate monobasic anhydrous ($NaH_2PO_4$) (BP329-1, Fisher Scientific) were used to prepare a 1 M stock sodium phosphate buffer solution containing 500 mM $Na_2HPO_4$ and 500 mM $NaH_2PO_4$ at an adjusted pH of 7.4. Sodium sulfate ($Na_2SO_4$) (239313, Sigma-Aldrich) was used to prepare a 1 M stock sodium sulfate buffer solution at an adjusted pH of 7.4. PFA (P6801, Sigma-Aldrich) and BAY 11-7082 (S2913, Selleckchem) were used as agents to elucidate underlying signal transduction mechanisms. Anti-IL6 (MP5-20F3, R&D Systems) and anti-IL1β (AF-401-NA, R&D Systems) were used as antibodies in a cell-based assay to neutralize the biological activity of targeted cytokines.

**Table 4.** Composition of control and adenine diets.

| Diet | Adenine (g/kg) | Available Pi (%) | Total Ca (%) | Protein source | Energy source | Pi source |
|---|---|---|---|---|---|---|
| TD.170303 (control diet) | 0 | 0.9 | 0.6 | Casein | 20% protein 66.9% carbs 13.2% fat | Casein Ca Pi, dibasic Na Pi, dibasic |
| TD.170304 (0.15% adenine) | 1.5 | 0.9 | 0.6 | Casein | 20% protein 66.8% carbs 13.2% fat | Casein Ca Pi, dibasic Na Pi, dibasic |
| TD.140290 (0.2% adenine) | 2 | 0.9 | 0.6 | Casein | 20% protein 66.8% carbs 13.2% fat | Casein Ca Pi, dibasic Na Pi, dibasic |

Pi, phosphate; Ca, calcium; Na, sodium. These diets were manufactured by Envigo.

## Mice

Animal studies were performed in the conformity with applicable laws and guidelines and were approved by the Animal Research Committee at the University of Alabama Birmingham School of Medicine (UAB). Studies were performed using male mice and were maintained on a NIH 31 rodent diet (Harlan Teklad) and fed ad libitum, unless otherwise indicated. Constitutive FGF Receptor four null (*Fgfr4*−/−) mice (*Weinstein et al., 1998*) were maintained on a C57BL/6 background. Constitutive *Col4a3*−/− null (Alport) mice (*Cosgrove et al., 1996*) were maintained on a mixed Sv129/C57BL/6 background. Both mouse models were housed in our UAB rodent facility, in a heterozygous breeding state.

For experiments exploring the contribution of pathologic FGF23-FGFR4 signaling to CKD-associated pathologies, 10- to 14-week-old *Fgfr4*−/− mice and corresponding wild-type littermates were placed on a customized diet containing 0.2% adenine (TD.140290, Envigo) for 6 weeks, switched to a customized diet containing 0.15% adenine (TD.170304, Envigo) for 2 weeks and transitioned back to the customized 0.2% adenine diet for an additional 6 weeks. Wild-type littermates placed on a customized control diet (TD.170303, Envigo) served as controls. All experimental groups were permitted a 1-week dietary acclimation period, using the customized control diet. After dietary acclimation, mice were unbiasedly assigned to either the customized control diet or customized adenine diet. After the 14-week duration, mice were euthanized under 2.5% isoflurane anesthesia and samples were prepared as described below. This experimental timeline is in accordance with previous studies (*Noonan et al., 2020*; *Taylor et al., 2019*). Adenine is known to induce kidney tubule-interstitial damage and is considered a dietary model of CKD.

For experiments testing the contribution of pathologic FGF23-FGFR4 signaling to systemic effects of a graded dietary phosphate load, 10- to 14-week-old *Fgfr4*−/− mice and corresponding wild-type littermates were unbiasedly assigned and fed a customized 0.7% phosphate diet (TD.180287, Envigo), a customized 2.0% phosphate diet (TD.08020, Envigo) or a customized 3.0% phosphate diet (TD.180286, Envigo) for 12 weeks. Wild-type littermates placed on the customized 0.7% phosphate

**Table 5.** Composition of 0.7%, 2%, and 3% phosphate (Pi) diets.

| Diet | Available Pi (%) | Total Ca (%) | Total iron (ppm) | Total K (%) | Total Na (%) | Protein source | Energy source | Pi source |
|---|---|---|---|---|---|---|---|---|
| TD.180287 (0.7% Pi diet) | 0.7 | 1.9 | 280 | 2.4 | 1.2 | Crude | 33.3% protein 53.9% carbs 12.8% fat | Crude protein |
| TD.08020 (2% Pi diet) | 2.0 | 1.9 | 280 | 1.8 | 0.9 | Crude | 33.3% protein 53.9% carbs 12.8% fat | Crude protein K Pi, monobasic Na Pi, monobasic |
| TD.180286 (3% Pi diet) | 3.0 | 1.9 | 280 | 2.4 | 1.2 | Crude | 33.3% protein 53.9% carbs 12.8% fat | Crude protein K Pi, monobasic Na Pi, monobasic |

Pi, phosphate; Ca, calcium; K, potassium; Na, sodium. These diets were manufactured by Envigo.

**Table 6.** Composition of 0.6% and 0.2% phosphate (Pi) diets.

| Diet | Available Pi (%) | Total Ca (%) | Total iron (ppm) | Total K (%) | Total na (%) | Protein source | Energy source | Pi source |
|---|---|---|---|---|---|---|---|---|
| TD.200407 (0.6% Pi diet, normal) | 0.6 | 0.6 | 40 | 0.6 | 0.38 | Egg white solids | 17.7% protein 65% carbs 17.3% fat | Egg white solids Ca Pi, monobasic |
| TD.200406 (0.2% Pi diet) | 0.2 | 0.6 | 40 | 0.6 | 0.38 | Egg white solids | 17.5% protein 65.4% carbs 17.1% fat | Egg white solids Ca Pi, monobasic |

Pi, phosphate; Ca, calcium; K, potassium; Na, sodium. These diets were manufactured by Envigo.

diet served as controls. At end of the experimental period, mice were euthanized under 2.5% isoflurane anesthesia and samples were prepared as described below.

For experiments investigating the contribution of hyperphosphatemia to CKD-associated pathologies, 4-week-old Alport mice and corresponding wild-type littermates were unbiasedly assigned and fed a customized 0.6% phosphate diet (TD.200407, Envigo) or a customized 0.2% phosphate diet (TD.200406, Envigo) as treatment for 6 weeks. Constitutive *Col4a3*$^{-/-}$ null mice are considered a genetic model of Alport syndrome and progressive CKD. When maintained on a mixed Sv129/C57BL/6 background, Alport mice die at 10 weeks of age due to rapid renal injury. Wild-type littermates placed on the customized 0.6% phosphate diet served as controls. At 10 weeks of age, mice were euthanized under 2.5% isoflurane anesthesia and samples were prepared as described below. A detailed description of diet compositions is indicated in *Table 4*; *Table 5*; *Table 6*. All experimental group numbers were predetermined on the basis of experience from previous publications. Investigators were not blinded to mouse genotypes.

## Serum chemistry

Mouse blood was collected by cardiac puncture and transferred into microvette serum gel tubes (20.1344, Sarstedt). Samples were then centrifuged at 10,000 × *g* for 5 min at room temperature. Serum supernatants were harvested and stored at −80°C. Clinical chemistry analyses were performed by the Animal Histopathology & Laboratory Medicine Core at the University of North Carolina, which is supported in part by an NCI Center Core Support Grant (5P30CA016086-41) to the UNC Lineberger Comprehensive Cancer Center. Serum intact FGF23 was assessed using ELISA (60-6800, Quidel).

## Grip-strength test

Muscle strength was assessed using a Chatillon DFE series digital force gauge (E-DFE-200, Chatillon) with a metal grid adaptor, provided by the Behavioral Assessment Core at UAB. Mice were allowed to grip the metal grid with fore- and hindlimbs and then gently pulled backwards by their tail, until mice could not grip the metal grid. Each mouse was given 10 trials, excluding the highest and lowest values. These eight trials were then averaged. These averaged values are used to represent the muscle grip strength of each individual mouse. Investigators were blinded to each experimental group.

## Mouse tissue collection

Unless otherwise indicated, tissues were excised, weighed, and either immediately flash frozen in liquid nitrogen or fixed for histologic examination. Organ and gastrocnemius weights were measured with an OHAUS scout portable balance (SJX323N/E).

## Histology

Spleen, kidney and gastrocnemius tissues were fixed in 10% formalin solution for 24 hr, transferred into 70% ethanol and subjected to paraffin embedding (IDEXX). Spleen, kidney, and gastrocnemius sections were cut and either stained with Perls' Prussian blue, H&E, or Masson's trichrome (IDEXX) and used for representative images. Images were captured on a Keyence BZ-X800 fluorescent microscope with a ×20 and ×40 objective lens.

## Tissue phosphate quantifications

To quantify liver phosphate concentrations in mouse tissue, liver samples were weighed, homogenized in protein precipitation solution (0.53 N HCl, 5.3% trichloroacetic acid [TCA]), boiled for 30 min at 95°C and cooled in room temperature water for 2 min. Samples were then centrifuged at 13,300 × *g* for 30 min at 4°C. Supernatants were collected and subjected to colorimetric phosphate quantifications (ab65622, Abcam) according to the manufactures' instructions.

## Tissue iron quantifications

To quantify nonheme iron concentrations in mouse tissues, spleen, and liver samples were weighed, homogenized in protein precipitation solution (0.53 N HCl, 5.3% TCA), boiled for 30 min at 95°C and cooled in room temperature water for 2 min. Samples were then centrifuged at 13,300 × *g* for 10 min at room temperature. Supernatants were harvested and subjected to colorimetric iron quantifications (157-30, Sekisui Diagnostics) according to the manufactures' instructions.

## Measurement of hematologic parameters and iron levels

Mouse blood was collected by cardiac puncture, transferred into microvette EDTA tubes (20.1341, Sarstedt), inverted to prevent clotting and stored at 4°C prior to shipment. Complete blood counts were measured by the Animal Histopathology & Laboratory Medicine Core at the University of North Carolina. In addition, serum supernatants were analyzed for iron- and total iron-binding capacity (TIBC) concentrations. Transferrin saturation percentage (TSAT%) = (serum iron/TIBC) × 100.

## Isolation and cultivation of mouse primary hepatocytes

Hepatocytes were isolated from 10- to 14-week-old male wild-type C57BL/6J mice, which were anesthetized and placed on a 37°C heated surface to maintain adequate body temperature. Ventral laparotomy from the pubis to the cranial border of the liver was performed and the abdominal wall was incised to both sides, caudal of the diaphragm, exposing the inferior vena cava (IVC). Following suprahepatic diaphragm incision and surgical silk (5/0) ligation of the thoracic IVC, the infrarenal IVC was cannulated using a 24-Gauge shielded catheter (381412, BD) attached to a perfusion line. A peristaltic pump was utilized to perfuse the liver with 30 ml of liver perfusion medium (17701-038, Gibco) followed by 30 ml of liver digest medium (17703-034, Gibco), both prewarmed in a 37°C water bath. The portal vein was incised to route consecutive retrograde perfusion through the liver at a rate of 3 ml/min until each solution was empty. Next, the digested liver was excised, transferred into a 10-cm dish containing hepatocyte wash medium (17704-024, Gibco) and was minced within a cell culture hood. The mixture was then filtered through a 70-µm nylon cell strainer (352350, Falcon) using a 20-ml plastic serological pipette into a 50-ml polypropylene conical tube (352098, Falcon) to remove debris. Cells were washed twice with chilled hepatocyte wash medium with centrifugation at 60 × *g* for 3 min at 4°C to allow a soft separation of parenchymal cells from nonparenchymal cells. To enrich the hepatocyte cell population, the cell pellet was resuspended and inverted four times in 20 ml of chilled 36% iso-osmotic percoll gradient solution (P1644, Sigma-Aldrich) (percoll gradient solution:William's E medium solution [four parts:six parts]) and centrifuged at 200 × *g* for 7 min at 4°C. The enriched hepatocyte population was resuspended in 10 ml of chilled hepatocyte wash medium and subjected to two washes with centrifugation at 60 × *g* for 2 min at 4°C. The washed pellet was resuspended in 12 ml of warm William's E medium (12551-032, Gibco) supplemented with primary hepatocyte thawing and plating supplements (CM3000, Gibco), counted in a hemocytometer after staining with trypan blue (25,900 Cl, Corning), seeded at a density of $2.5 \times 10^5$ cells/6-well or $1.0 \times 10^5$ cells/12-well on plates coated with 100 µg/ml of collagen type 1 (354236, Corning) and allowed to adhere for 4 hr in a humidified 5% $CO_2$ incubator at 37°C. After this attachment period, medium was exchanged with fresh warm William's E medium solution supplemented with primary hepatocyte maintenance supplements (CM4000, Gibco) and incubated overnight in a humidified 5% $CO_2$ incubator at 37°C. Next morning, media was exchanged with fresh warm Dulbecco's Modified Eagle's Medium (DMEM) (26140079, Gibco) supplemented with 1× penicillin/streptomycin (15140122, Gibco) and incubated for 6 hr in a humidified 5% $CO_2$ incubator at 37°C. This 6 hr serum-starvation period allows cells to synchronize to an identical cell cycle arrest phase, thus eliminating the potential impact between contrasting cell cycles and a cells overall response to exogenous treatment, as serum contains various

growth factors and cytokines which promote the activation of signal transduction pathways related to cell proliferation and survival.

## Cell culture

Hepatocytes were isolated, cultivated, and serum starved as described in supplemental methods. For experiments investigating the activation of signal transduction mediators, cells were seeded on 6-well collagen-coated plates and treated with either TNFα (100 ng/ml), IL6 (50 ng/ml), FGF23 (25 ng/ml), or appropriate amounts of sodium phosphate (1 M; pH 7.4) and sodium sulfate (1 M; pH 7.4) buffers to produce final desired concentrations and incubated for 30 min in a humidified 5% $CO_2$ incubator at 37°C. DMEM supplemented with 1× penicillin/streptomycin, which contains ~1 mM phosphate, served as a reference control (Ctrl). Sodium sulfate served as a negative control in response to increased anions.

For experiments analyzing expression levels of specific target genes, cells were seeded on 12-well collagen-coated plates and treated with either LPS (100 ng/ml), IL6 (50 ng/ml), or appropriate amounts of sodium phosphate (1 M; pH 7.4) and sodium sulfate (1 M; pH 7.4) buffers to produce final desired concentrations and incubated for 24 hr in a humidified 5% $CO_2$ incubator at 37°C. As described above, DMEM with 1× penicillin/streptomycin served as a reference control (Ctrl) and sodium sulfate served as a negative control.

For experiments investigating the role of high extracellular phosphate, cells were seeded on either 6- or 12-well collagen-coated plates and preincubated for 1 hr with or without the addition of PFA (1 mM) in a humidified 5% $CO_2$ incubator at 37°C. Cells were then either treated for 30 min to assess NFκB activation or treated for 24 hr to analyze expression levels of specific target genes, and incubated accordingly in a humidified 5% $CO_2$ incubator at 37°C. Specific treatments were conducted with factors described above. DMEM with 1× penicillin/streptomycin served as a reference control (Ctrl).

For experiments analyzing the participation of NFκB signaling, cells were seeded on 12-well collagen-coated plates and preincubated for 1 hr with or without the addition of BAY 11-7082 (20 μM) in a humidified 5% $CO_2$ incubator at 37°C. Cells were then treated and incubated for 24 hr in a humidified 5% $CO_2$ incubator at 37°C to analyze expression levels of specific target genes. Specific treatments were conducted with factors described above. DMEM with 1× penicillin/streptomycin served as a reference control (Ctrl). Total protein lysates were prepared from 30-min treatments as described below. Total RNA was prepared from 24-hr treatments as described below. All 24-hr treatments were supplemented with 0.70% fetal bovine serum (FBS) (CM3000, Gibco).

## Cytokine neutralization

Hepatocytes were seeded on 12-well collagen-coated plates, cultivated and serum starved as described in supplemental methods. Primary mouse hepatocytes were treated for 24 hr to analyze expression levels of specific target genes with either LPS (100 ng/ml) or appropriate amounts of sodium phosphate buffer (1 M; pH 7.4) to produce a final desired phosphate concentration and incubated accordingly in a humidified 5% $CO_2$ incubator at 37°C. Treatments were performed with or without the addition of neutralizing antibodies against IL6 (6 μg/ml) and/or IL1β (6 μg/ml) as indicated. Total RNA was prepared from treatments as described below. All treatments were supplemented with 0.70% FBS (CM3000, Gibco).

## RNA isolation and quantification

Total RNA was extracted from liver and cultured hepatocytes using a RNeasy Plus Mini Kit (74136, Qiagen) and from gastrocnemius tissue using a RNeasy Plus Universal Mini Kit (73404, Qiagen) following the manufactures' instructions. Employing a two-step reaction method, 1 μg of total RNA was reverse transcribed into cDNA using iScript Reverse Transcription Supermix (1708840, BioRad). Quantitative PCR was performed with 100 ng of cDNA, SsoAdvanced Universal SYBR Green Supermix (172-5272, BioRad) and sequence specific primers (as indicated in *Table 7*). Samples were run in duplicate on a CFX96 Touch Real-Time Detection Instrument (1855196, BioRad). Amplification was performed in forty cycles (95°C, 30 s; 98°C, 15 s; 60°C, 30 s; 65°C, 5 s). The generated amplicon was systematically double checked by its melting curve. Relative gene expression was normalized to expression levels of housekeeping genes *18S rRNA* (for in vitro studies) or *Gapdh* (for in vivo studies).

**Table 7.** Oligonucleotides used as sequence specific primers in quantitative polymerase chain reaction (qPCR) analyses.

| Gene | Species | Orientation | Primer sequence (5'–3') |
|------|---------|-------------|-------------------------|
| Npt-1/Slc17a1 | Mus musculus | Forward<br>Reverse | GGC ACC TCC CTT AGA ACG AG<br>CAG AAC ACA CCC AAC AAT ACC AAA |
| Npt-4/Slc17a3 | Mus musculus | Forward<br>Reverse | TGG TAC CCA TTG TTG CTG GC<br>GGG ACA GCT TCA CAA ACG AGT |
| NaPi-2a/Slc34a1 | Mus musculus | Forward<br>Reverse | TCA TTG TCA GCA TGG TCT CCT C<br>CCT GCA AAA GCC CGC CTG |
| NaPi-2b/Slc34a2 | Mus musculus | Forward<br>Reverse | CTC CTG CTG TCC CTT ACC TG<br>TGT CAT TTG TTT TGC TGG CCT C |
| NaPi-2c/Slc34a3 | Mus musculus | Forward<br>Reverse | GAT GCC TTT GAC CTG GTG GA<br>GCC ATG CCA ACC TCT TTC AG |
| PiT-1/Slc20a1 | Mus musculus | Forward<br>Reverse | TTC CTT GTT CGT GCG TTC ATC<br>AAT TGG TAA AGC TCG TAA GCC ATT |
| PiT-2/Slc20a2 | Mus musculus | Forward<br>Reverse | GAC CGT GGA AAC GCT AAT GG<br>CTC AGG AAG GAC GCG ATC AA |
| Fgfr1 | Mus musculus | Forward<br>Reverse | GCT TGA CGT CGT GGA ACG AT<br>AGC CAC TGA ATG TGA GGC TG |
| Fgfr2 | Mus musculus | Forward<br>Reverse | ATC CCC CTG CGG AGA CA<br>GAG GAC AGA CGC GTT GTT ATC C |
| Fgfr3 | Mus musculus | Forward<br>Reverse | GTG TGC GTG TAA CAG ATG CTC<br>CGG GCG AGT CCA ATA AGG AG |
| Fgfr4 | Mus musculus | Forward<br>Reverse | TGA AGA GTA CCT TGA CCT CCG<br>TCA TGT CGT CTG CGA GTC AG |
| Alt1/Gpt1 | Mus musculus | Forward<br>Reverse | GCC CTC GAG TAC TAT GCG TC<br>TGT CTT GGT ATA CCT CAT CAG CC |
| Ast1/Got1 | Mus musculus | Forward<br>Reverse | CTG AAT GAT CTG GAG AAT GCC C<br>TGC AAA GCC CTG ATA GGC TG |
| Il6 | Mus musculus | Forward<br>Reverse | CTC TGG GAA ATC GTG GAA AT<br>CCA GTT TGG TAG CAT CCA TC |
| Il1b | Mus musculus | Forward<br>Reverse | TGC CAC CTT TTG ACA GTG ATG<br>TGA TGT GCT GCT GCG AGA TT |
| Saa1 | Mus musculus | Forward<br>Reverse | ACA CCA GCA GGA TGA AGC TAC T<br>GAG CAT GGA AGT ATT TGT CTG AGT |
| Hamp | Mus musculus | Forward<br>Reverse | GAG CAG CAC CAC CTA TCT CC<br>TTG GTA TCG CAA TGT CTG CC |
| Haptoglobin/Hp | Mus musculus | Forward<br>Reverse | AGA GAG GCA AGA GAG GTC CA<br>GGC AGC TGT CAT CTT CAA AGT |
| Atrogin1/Fbxo32 | Mus musculus | Forward<br>Reverse | TGA GCG ACC TCA GCA GTT AC<br>GCG CTC CTT CGT ACT TCC TT |
| Murf1/Trim63 | Mus musculus | Forward<br>Reverse | GAG GGC CAT TGA CTT TGG GA<br>TGG TGT TCT TCT TTA CCC TCT GT |
| Mstn | Mus musculus | Forward<br>Reverse | CTC CAG AAT AGA AGC CAT A<br>GCA GAA GTT GTC TTA TAG C |
| Mt1 | Mus musculus | Forward<br>Reverse | CGA CTT CAA CGT CCT GAG TAC<br>AGG AGC TGG TGC AAG TG |
| 18S rRNA/Rn18s | Mus musculus | Forward<br>Reverse | TTG ACG GAA GGG CAC CAC CAG<br>GCA CCA CCA CCC ACG GAA TCG |
| Gapdh | Mus musculus | Forward<br>Reverse | CCA ATG TGT CCG TCG TGG ATC T<br>GTT GAA GTC GCA GGA GAC AAC C |

Results were evaluated using the $2^{-\Delta\Delta Ct}$ method and expressed as mean ± standard error of the mean (SEM).

## Protein isolation and immunoblotting

Total protein was extracted from cells which were placed on ice and scraped from 6- or 12-well plates, using a 300 or 150 µl volume of RIPA lysis buffer (50 mM Tris–HCl, pH 7.5, 200 mM NaCl, 1% Triton X-100, 0.25% deoxycholic acid, 1 mM EDTA, 1 mM ethylene glycol-bis(β-aminoethyl ether)-N,N,N',N'-tetraacetic acid tetrasodium (EGTA)), respectively, with addition of protease inhibitor (11836153001, Roche) and phosphatase inhibitors (P5726, P0044, Sigma-Aldrich). Cell lysates were then incubated on ice for 30 min and cleared by centrifugation at 13,000 × $g$ for 30 min at 4°C. Supernatants were collected and protein was quantified using a Pierce BCA Protein Assay Kit (23225, Thermo Fisher Scientific).

Following protein quantification, supernatants were appropriately aliquoted and suspended in volumes of Laemmli sample buffer (1610747, BioRad) with β-mercaptoethanol (1610710, BioRad) as reducing agent, denatured at 100°C for 5 min and stored at −80°C. Protein samples [20 µg total protein] were loaded onto 8% or 10% SDS polyacrylamide gels and separated by SDS–PAGE. Poly-acrylamide gels were run in 1× Tris/Glycine/SDS buffer (1610732, BioRad) at 20 mA per gel and stopped when sample dyes reached the end of the gels. Proteins were electroblotted onto PVDF membranes (IPVH00010, Merck Millipore) via a semi-dry cassette (1703940, BioRad) in 1× Tris/Glycine Buffer (1610734, BioRad) with 20% methanol at 20 V for 1 hr. Membranes were then blocked in 5% nonfat dry milk with 0.1% Tween-20 diluted in 1× Tris buffered saline (TBS) pH 7.5 for 1 hr and probed with primary antibodies at 1:1000 against specific antigens overnight at 4°C. ERK1/2 (4695, Cell Signaling), STAT3 (4904, Cell Signaling), NFκB (8242, Cell Signaling), and β-actin (4970, Cell Signaling) primary antibodies were used in 1× TBS with 5% nonfat dry milk and 0.1% Tween-20. Phos-pho-ERK1/2 (9101, Cell Signaling), phospho-STAT3 (9145, Cell Signaling), and phospho-NFκB (3033, Cell Signaling) primary antibodies were used in 1× TBS with 5% BSA and 0.1% Tween-20.

Next day, membranes were subjected to three wash periods for 5 min in 1× TBS with 0.1% Tween and then probed with horseradish peroxidase-conjugated goat anti-mouse or goat anti-rabbit secondary antibodies at 1:2,500 (W4021, W4011, Promega) in 1× TBS with 5% nonfat dry milk and 0.1% Tween at room temperature for 1 hr. Membranes were then subjected to three wash periods for 10 min in 1× TBS with 0.1% Tween at room temperature. Horseradish peroxidase activity was detected using enhanced chemiluminescence detection solution (RPN2106, GE Healthcare) and imaged on an SRX-101A X-ray film developer. All immunoblots were repeated with a minimum of three independent trials, with comparable results.

## Statistics

Data organization, scientific graphing, and statistical significance of differences between experimental groups were performed by using GraphPad Prism (version 9.0.0). All results are expressed as mean ± SEM. Depending on number of experimental groups and factors analyzed, we performed a two-way analysis of variance (ANOVA) followed by a post hoc Tukey test (for studies affected by two factors) or in the form of a one-way ANOVA (for studies measuring variance in three groups or more). Correlation and slope analyses were examined by simple linear regression. Statistical significance was set at a p value of less than or equal to 0.05. Sample size was determined on the basis of sample avail-ability, prior experimental studies performed in our laboratory and from prior literature. No formal randomization was used in any experiment. For in vivo experiments, animals were unbiasedly assigned into different experimental groups, regardless of genotype. Group allocation was not performed in a blinded manner. Whenever possible, investigators were blinded to experimental groups (e.g., analysis of all grip-strength measurements).

## Study approval

All animal protocols and experimental procedures for adenine diet in $FGFR4^{+/+}$ and $Fgfr4^{-/-}$ mice, graded phosphate diets in $Fgfr4^{+/+}$ and $Fgfr4^{-/-}$ mice, low phosphate diets in $Col4a3^{+/+}$ and $Col4a3^{-/-}$ mice and primary hepatocyte isolations from wild-type C57BL/6J mice, were approved by the Institutional Animal Care and Use Committees (IACUC) at the University of Alabama Birmingham School of Medi-cine. All animals were maintained in a ventilated rodent-housing system with temperature-controlled

environments (22–23°C) with a 12-hr light/dark cycle and allowed ad libitum access to food and water. All protocols adhered to the Guide for Care and Use of Laboratory Animals to minimize pain and suffering. No animals were excluded from analysis.

## Acknowledgements

BC designed and performed experiments, analyzed data, and wrote the manuscript. KH, IC, CY, DK, and DW assisted with experiments. GJ measured liver and spleen iron concentrations in Alport mice. OG, JBL, IBS, and MH assisted with data analysis and interpretation. CF supervised project, secured funding, assisted with data interpretation, and edited manuscript. All authors discussed results, read, and contributed edits to manuscript and approved final version.

This study was supported by NIH grants F31-DK-117550 (BC), T90-DE-022736 (KH), F31-DK-115074 (CY), K24-DK-116180 (OG), R01-DK-087727 (JLB), U01-DK-119950 (IBS), K08-DK-111980 (MH), R01-HL-128714, R01-HL-145528, and R01-DK-125459 (CF); and by grants from the Deutsche Forschungsgemeinschaft (DK) and the National Science Foundation (IC). Furthermore, CF was supported by the UAB-UCSD O'Brien Core Center for Acute Kidney Injury Research, the AMC21 program of the Department of Medicine at UAB and the Tolwani Innovation Award from the Division of Nephrology at UAB; JLB was supported by the Patricia and Scott Eston Massachusetts General Hospital Research Scholar Award.

## Additional information

### Competing interests

Orlando Gutierrez: has received honoraria and grant support from Akebia and Amgen, grant support from GSK, honoraria from Ardelyx, Reata, and AstraZeneca, and serves on the Data Monitoring Committee for QED. Jodie L Babitt: has ownership interest in Ferrumax Pharmaceuticals and has been a consultant for Incyte Corporation, and Alnylam Pharmaceuticals. Christian Faul: has served as a consultant for Bayer and Calico Labs, and he is the founder and currently the CSO of a startup biotech company (Alpha Young LLC). The other authors declare that no competing interests exist.

### Funding

| Funder | Grant reference number | Author |
| --- | --- | --- |
| National Institutes of Health | F31-DK-117550 | Brian Czaya |
| National Institutes of Health | F31-DK-115074 | Christopher Yanucil |
| National Institutes of Health | R01-HL-128714 | Christian Faul |
| National Science Foundation | Alabama Louis Stoke Alliance of Minority Participation (LSAMP) Bridge to the Doctorate (BD) | Isaac Campos |
| Deutsche Forschungsgemeinschaft | Research Fellowship | Dominik Kentrup |
| National Institutes of Health | R01-HL-145528 | Christopher Yanucil |
| National Institutes of Health | R01-DK-125459 | Christian Faul |

The funders had no role in study design, data collection, and interpretation, or the decision to submit the work for publication.

## Author contributions
Brian Czaya, Conceptualization, Data curation, Formal analysis, Investigation, Methodology, Writing – original draft; Kylie Heitman, Isaac Campos, Christopher Yanucil, Grace Jung, Investigation, Methodology; Dominik Kentrup, Formal analysis, Methodology; David Westbrook, Project administration; Orlando Gutierrez, Conceptualization; Jodie L Babitt, Conceptualization, Methodology; Isidro B Salusky, Conceptualization, Investigation; Mark Hanudel, Conceptualization, Investigation, Methodology; Christian Faul, Conceptualization, Funding acquisition, Investigation, Project administration, Resources, Supervision, Writing – original draft

## Author ORCIDs
Brian Czaya ![ORCID] http://orcid.org/0000-0002-7888-2432
Kylie Heitman ![ORCID] http://orcid.org/0000-0002-5345-2727
Christian Faul ![ORCID] http://orcid.org/0000-0002-7512-0977

## Ethics
All animal protocols and experimental procedures for adenine diet in FGFR4+/+ and FGFR4−/− mice, graded phosphate diets in FGFR4+/+ and FGFR4−/− mice, low phosphate diets in COL4A3+/+ and COL4A3−/− mice and primary hepatocyte isolations from wild-type C57BL/6J mice, were approved by the Institutional Animal Care and Use Committees (IACUC) at the University of Alabama Birmingham School of Medicine (#22089). All animals were maintained in a ventilated rodent-housing system with temperature-controlled environments (22–23°C) with a 12-hr light/dark cycle and allowed ad libitum access to food and water. All protocols adhered to the Guide for Care and Use of Laboratory Animals to minimize pain and suffering. No animals were excluded from analysis.

## Decision letter and Author response
Decision letter https://doi.org/10.7554/eLife.74782.sa1
Author response https://doi.org/10.7554/eLife.74782.sa2

# Additional files

## Supplementary files
• Transparent reporting form

## Data availability
All data generated and analyzed during this study are available through Dryad.

The following dataset was generated:

| Author(s) | Year | Dataset title | Dataset URL | Database and Identifier |
|---|---|---|---|---|
| Faul C | 2022 | Data from: Hyperphosphatemia increases inflammation to exacerbate anemia and skeletal muscle wasting independently of FGF23-FGFR4 signaling full source data | https://dx.doi.org/10.5061/dryad.6t1g1jx0f | Dryad Digital Repository, 10.5061/dryad.6t1g1jx0f |

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
