## [Editor Report]

Many of us have followed the work of your group and others on FGF23 excess and hyperphosphatemia and the morbidity of chronic kidney disease in both animal models and in the human disease state. Inflammation, anemia and muscle wasting are clearcut serious consequences to deal with CKD. Animal models that will allow us to further elucidate the signaling and effector pathways activated by hyperphosphatemia are welcome advances in the field.

---

## [Decision Letter]

**Decision letter after peer review:**

Thank you for submitting your article "Hyperphosphatemia increases inflammation to exacerbate anemia and skeletal muscle wasting independently of FGF23-FGFR4 signaling" for consideration by *eLife*. Your article has been reviewed by 2 peer reviewers, and the evaluation has been overseen by a Reviewing Editor and Mone Zaidi as the Senior Editor. The following individuals involved in review of your submission have agreed to reveal their identity: Seiji Fukumoto (Reviewer #1); Mohammed S Razzaque (Reviewer #3).

Please see the reviews listed below.

We feel the essential things to revise are the following.

From Reviewer #1:

1. It is not clearly shown why model mice of Alport syndrome rather than adenine-induced CKD model were used for examining effects of low phosphate diet.

2. It is difficult to see where iron is deposited by the figures. Higher magnification figures are necessary.

3. It is not clear where the supplemental Materials and methods are (l. 566, 576, 585….).

4. Histology of muscle sections seems to be quite different between groups (For example Figure 3E). How were these sections selected?

From Reviewer #3:

1. Can you speak to this point please?

This manuscript is technically sound and addresses an important issue of the direct effect of excessive phosphate in inducing inflammation. The authors also tried to connect the high phosphate and skeletal muscle wasting or anemia to give relevance to chronic kidney disease. The authors also ruled out the involvement of FGF23-FGFR4 in iron deficiency by providing convincing evidence of mouse genetics.

2. Can you address these requests please?

The results are of interest and the readership will benefit from the following modifications/ clarifications:

A. The reviewer realizes that the heart is not the focus of this study, considering the authors have shown the involvement of FGF23-FGFR4 signaling in cardiac hypertrophy; commenting on the FGF23-FGFR4-independent effects of high phosphate on cardiac structure or function would be useful to know. In other words, do the high phosphate has a differential impact on cardiac vs. skeletal muscle?

B. For a high phosphate-induced downstream signaling cascade (Figure 6), the authors should consider explaining their results in the context of the global cytotoxic signaling cascade generated following excessive phosphate exposure (PMID: 32892444).

C. It would be nice to know how phosphate activates Pit1 and Pit2?

D. The part of the Discussion section is the repetition of the result section, and the authors should consider minimizing the overlap.

Can you please make the revisions and adjustments requested? We are enthusiastic about this paper. Thank you.

*Reviewer #1:*

The authors investigated the effects of phosphate on the development of several complications of chronic kidney disease (CKD) such as inflammation, anemia, and skeletal muscle wasting using several animal models. They showed that phosphate induced these complications in adenine-induced CKD model and mice fed with high phosphate diet in FGFR4 independent manner. They also showed that a low phosphate diet improved these in model mice of Alport syndrome. Then, they investigated the actions of phosphate using primary hepatocytes and indicated that phosphate induced the expression of several inflammatory cytokines. These data reveal the new function of phosphate and are potentially important for the understanding and the management of the complication of CKD.

Strengths

The authors used several established in vivo models of CKD. Data obtained in these in vivo experiments are clear and reproducible in multiple models.

They showed novel actions of phosphate in the development of complications of CKD.

Weaknesses

Several in vitro experiments did not completely reveal the underlying mechanism of the actions of phosphate. It is not clear how phosphate phosphorylates P65. The effects of phosphate on enhanced phosphorylation of P65 were not dose-dependent (Figure 6A). It is not shown whether phosphate affected the expression of TNFα while TNFα is a potent activator of NFκB (Figure 6A). It is not shown either whether IL1β had the same effects on the expressions of cytokines and hepcidine. A scheme summarizing in vitro results and proposing the mechanism of phosphate might be helpful.

Mechanistic investigations on muscle wasting are lacking.

1. It is not clearly shown why model mice of Alport syndrome rather than adenine-induced CKD model were used for examining effects of low phosphate diet.

2. It is difficult to see where iron is deposited by the figures. Higher magnification figures are necessary.

3. It is not clear where the supplemental Materials and methods are (l. 566, 576, 585….).

4. Histology of muscle sections seems to be quite different between groups (For example Figure 3E). How were these sections selected?

*Reviewer #3:*

This manuscript focuses on the impact of high phosphate on skeletal muscle injury and beyond. Although there are gaps, this manuscript would likely to form the basis for further in-depth studies.

This manuscript is technically sound and addresses an important issue of the direct effect of excessive phosphate in inducing inflammation. The authors also tried to connect the high phosphate and skeletal muscle wasting or anemia to give relevance to chronic kidney disease. The authors also ruled out the involvement of FGF23-FGFR4 in iron deficiency by providing convincing evidence of mouse genetics.

The results are of interest and the readership will benefit from the following modifications/ clarifications:

The reviewer realizes that the heart is not the focus of this study, considering the authors have shown the involvement of FGF23-FGFR4 signaling in cardiac hypertrophy; commenting on the FGF23-FGFR4-independent effects of high phosphate on cardiac structure or function would be useful to know. In other words, do the high phosphate has a differential impact on cardiac vs. skeletal muscle?

For a high phosphate-induced downstream signaling cascade (Figure 6), the authors should consider explaining their results in the context of the global cytotoxic signaling cascade generated following excessive phosphate exposure (PMID: 32892444).

It would be nice to know how phosphate activates Pit1 and Pit2?

The part of the Discussion section is the repetition of the result section, and the authors should consider minimizing the overlap.

---

## [Author Response]

Can you please make the revisions and adjustments requested? We are enthusiastic about this paper. Thank you.

We thank the Editors for being enthusiastic about our work, for giving us the opportunity to respond to the Reviewers’ insightful and constructive comments, and for inviting us to submit a revised manuscript. Our specific responses to the Reviewers' comments are presented below in point-by-point fashion. We now submit a revised manuscript with alterations, highlighted in yellow, and we hope that the revised manuscript is suitable for publication in *eLife*.

Reviewer #1:The authors investigated the effects of phosphate on the development of several complications of chronic kidney disease (CKD) such as inflammation, anemia, and skeletal muscle wasting using several animal models. They showed that phosphate induced these complications in adenine-induced CKD model and mice fed with high phosphate diet in FGFR4 independent manner. They also showed that a low phosphate diet improved these in model mice of Alport syndrome. Then, they investigated the actions of phosphate using primary hepatocytes and indicated that phosphate induced the expression of several inflammatory cytokines. These data reveal the new function of phosphate and are potentially important for the understanding and the management of the complication of CKD.

We thank the Reviewer for the positive feedback and for recognizing the novelty of our work.

StrengthsThe authors used several established in vivo models of CKD. Data obtained in these in vivo experiments are clear and reproducible in multiple models.They showed novel actions of phosphate in the development of complications of CKD.WeaknessesSeveral in vitro experiments did not completely reveal the underlying mechanism of the actions of phosphate. It is not clear how phosphate phosphorylates P65. The effects of phosphate on enhanced phosphorylation of P65 were not dose-dependent (Figure 6A). It is not shown whether phosphate affected the expression of TNFα while TNFα is a potent activator of NFκB (Figure 6A). It is not shown either whether IL1β had the same effects on the expressions of cytokines and hepcidine. A scheme summarizing in vitro results and proposing the mechanism of phosphate might be helpful.

We thank the Reviewer for this suggestion. We now present a schematic of the model and pathomechanism, including the signaling events in hepatocytes, in new Figure 8.

Mechanistic investigations on muscle wasting are lacking.

We agree with the Reviewer. We are currently investigating the exact impact on skeletal muscle, including the underlying pathomechanism in myotubes. However, we believe that this aspect is beyond the scope of the manuscript.

1. It is not clearly shown why model mice of Alport syndrome rather than adenine-induced CKD model were used for examining effects of low phosphate diet.

Adenine diet-induced nephropathy as a mouse model of CKD can be heterogeneous, ranging from a mild/no phenotype to a massive/lethal phenotype. Additionally, examining whether dietary phosphate restriction could represent a potential clinical treatment in this specific mouse model would be challenging, due to its phenotypic heterogeneity and subsequent diet already supplied. Designing a phosphate-restricted diet containing adenine, including proper control diets, would be challenging. After careful consideration, to investigate the potential of a low phosphate diet as clinical treatment, Alport mice were chosen. As an established genetic model of CKD, Alport mice provide a greater degree of phenotypic homogeneity, along with the development of endogenous hyperphosphatemia with severe inflammation, anemia and skeletal muscle wasting, overall making them an ideal candidate for dietary intervention.

2. It is difficult to see where iron is deposited by the figures. Higher magnification figures are necessary.

We have now included larger magnifications of spleen tissue sections stained with Perl’s Prussian blue from *Fgfr4^+/+^* and *Fgfr4^-/-^* mice fed either adenine diet or a graded dietary phosphate (Pi) load, as well as from wild-type (*Col4a3^+/+^*) and Alport (*Col4a3^-/-^*) mice fed either control diet (0.6% Pi) or a low phosphate diet (0.2% Pi) (new Figure1—figure supplement 1c, Figure 2—figure supplement 1f, Figure 4—figure supplement 1e)

3. It is not clear where the supplemental Materials and methods are (l. 566, 576, 585….).

We apologize and this was a mistake from our side. There are no supplemental Materials and methods, and all the information is within the Materials and methods section of the manuscript.

4. Histology of muscle sections seems to be quite different between groups (For example Figure 3E). How were these sections selected?

Between groups, all gastrocnemius sections were selected at random. Unfortunately, these alluded differences are a result from variations during the paraffin embedding process, which were performed by IDEXX Laboratories. Although embedding variations are noticeable, we think that the pathological differences in myofibers are apparent between groups.

Reviewer #3:This manuscript focuses on the impact of high phosphate on skeletal muscle injury and beyond. Although there are gaps, this manuscript would likely to form the basis for further in-depth studies.This manuscript is technically sound and addresses an important issue of the direct effect of excessive phosphate in inducing inflammation. The authors also tried to connect the high phosphate and skeletal muscle wasting or anemia to give relevance to chronic kidney disease. The authors also ruled out the involvement of FGF23-FGFR4 in iron deficiency by providing convincing evidence of mouse genetics.

We thank the Reviewer for the positive evaluation of our main findings. As pointed out, we address the effects of excessive phosphate on regulators of iron metabolism, iron status and skeletal muscle function. First, utilizing mice which harbor global FGFR4 deletion and exposing them to an adenine-rich diet to induce CKD, we show ablating FGF23-FGFR4 signaling does not protect mice from these CKD-associated complications. Supporting this observation, utilizing an established genetic model of progressive CKD, we subject Alport mice to a low phosphate diet as treatment. These observations provide evidence that restricting dietary phosphate leads to a ~ 5 to 10 mg/dL reduction in systemic phosphate levels in comparison to Alport mice on control diet (0.6% phosphate), and alleviates the degree of inflammation, anemia and skeletal muscle dysfunction.

The results are of interest and the readership will benefit from the following modifications/ clarifications:The reviewer realizes that the heart is not the focus of this study, considering the authors have shown the involvement of FGF23-FGFR4 signaling in cardiac hypertrophy; commenting on the FGF23-FGFR4-independent effects of high phosphate on cardiac structure or function would be useful to know. In other words, do the high phosphate has a differential impact on cardiac vs. skeletal muscle?

The Reviewer raises an interesting point. Current direct effects of phosphate on the myocardium are not known, and in our opinion, is also difficult to study since it would require animal models with heart-specific loss-of-function approaches for phosphate uptake. This would be challenging since (a) phosphate uptake in the heart might be mediated by more than one transporter, and (b) the complete blockade of phosphate uptake into cardiac cells would be most likely detrimental as all cells depend on constant phosphate uptake for various house-keeping functions, especially myocytes which demand a lot of energy. Cell culture studies suggest that phosphate might affect cardiac cells directly, including the induction of hypertrophic growth of cardiac myocytes (PMID: 25326585; 25999956), but in vivo proof is still missing. We have previously shown that the deletion or blockade of FGFR4 in animal models with high phosphate diet and normal kidney function as well as in animal models with CKD, protects the heart from hypertrophy (PMID: 26437603, 28512310). These findings suggest that FGF23/FGFR4, and not phosphate, is the main driver of cardiac hypertrophy in the context of hyperphosphatemia. It is also possible that FGFR4 acts as phosphate receptor (it has been recently shown that *FGFR1* can sense phosphate in bone cells; PMID: 31097591), but we believe that FGF23 is the main ligand for FGFR4.

For a high phosphate-induced downstream signaling cascade (Figure 6), the authors should consider explaining their results in the context of the global cytotoxic signaling cascade generated following excessive phosphate exposure (PMID: 32892444).

We appreciate this suggestion. We added additional information and citations to the Discussion section.

It would be nice to know how phosphate activates Pit1 and Pit2?

This is an important question. PiT_1/2_ can act as sensors as well as transporters for phosphate, as shown in other cell types. Other investigators currently focus on this topic and try to determine if PiT_1/2_ can bind phosphate which induces structural alterations that lead to changes in signal transduction and/or mediate phosphate uptake so that intracellular sensors detect phosphate and alter signaling events. Studies in this area are still early and mainly done with overexpressed recombinant proteins. We have also started to look into these mechanistic details, but these studies are still early and we believe this is also beyond the scope of the current manuscript.

The part of the Discussion section is the repetition of the result section, and the authors should consider minimizing the overlap.

We appreciate this suggestion. In order to accommodate readers who might only read specific sections of the manuscript, such as the discussion instead of the results, we would prefer to retain the level of detail in our Discussion section. Additionally, this same level of detail could assist readers who might reside in other research fields, that only read the discussion. We believe this brief overlap combined with its interpretations will have a more significant impact on the broad implications for these specific readers.